# Creating a bacterium that forms eukaryotic nucleosome core particles

Xinyun Jing[1,8], Niubing Zhang[1,2,8], Xiaojuan Zhou[1,3,8], Ping Chen[1,4,8], Jie Gong[1,3,8], Kaixiang Zhang[1,2,3], Xueting Wu[1], Wenjuan Cai[5], Bang-Ce Ye [2], Pei Hao [3,4], Guo-ping Zhao [1,6,7], Sheng Yang [1,3] & Xuan Li [1,3] ✉

The nucleosome is one of the hallmarks of eukaryotes, a dynamic platform that supports many critical functions in eukaryotic cells. Here, we engineer the in vivo assembly of the nucleosome core in the model bacterium *Escherichia coli*. We show that bacterial chromosome DNA and eukaryotic histones can assemble in vivo to form nucleosome complexes with many features resembling those found in eukaryotes. The formation of nucleosomes in *E. coli* was visualized with atomic force microscopy and using tripartite split green fluorescent protein. Under a condition that moderate histones expression was induced at 1 μM IPTG, the nucleosome-forming bacterium is viable and has sustained growth for at least 110 divisions in longer-term growth experiments. It exhibits stable nucleosome formation, a consistent transcriptome across passages, and reduced growth fitness under stress conditions. In particular, the nucleosome arrays in *E. coli* genic regions have profiles resembling those in eukaryotic cells. The observed compatibility between the eukaryotic nucleosome and the bacterial chromosome machinery may reflect a prerequisite for bacteria-archaea union, providing insight into eukaryogenesis and the origin of the nucleosome.

The nucleosome is the basic structural unit in the packaging of genomic DNA in eukaryotic cells. As one of the hallmarks of eukaryotes, the nucleosome acts as a dynamic platform supporting many essential functions, such as transcription regulation, DNA replication and repair, mitotic division, and epigenetic modulation[1–3]. The nucleosome core contains a histone octamer consisting of two copies of each of four histones, H2A, H2B, H3, and H4, and ~147 bp of double helix DNA tightly wrapped around the octamer in ~1.7 turns[4,5]. The H1 linker histone is not part of the nucleosome core but binds to DNA between neighboring nucleosomes, serving as a linker molecule in the formation of chromatin fibers, the higher-order structure of eukaryotic chromosomes[6,7].

The in vivo assembly of the nucleosome core in eukaryotes is a stepwise process initiated by the joining of two H3/H4 dimers to form a tetrasome partially wrapped by a DNA segment. Subsequently, two H2A/H2B dimers are individually deposited, leading to the formation of a complete nucleosome core, the octasome[8,9]. Nucleosome complexes are dynamic, and their assembly and disassembly involve multiple transient intermediate states and interactions of many different components[1,10,11]. In eukaryotic cells, nucleosome positioning,

[1]Key Laboratory of Synthetic Biology, Key Laboratory of Plant Design, CAS Center for Excellence in Molecular Plant Sciences, Chinese Academy of Sciences, Shanghai, China. [2]State Key Laboratory of Bioreactor Engineering, East China University of Science and Technology, Shanghai, China. [3]University of Chinese Academy of Sciences, Beijing, China. [4]Key Laboratory of Molecular Virology and Immunology, Shanghai Institute of Immunity and Infection, Chinese Academy of Sciences, Shanghai, China. [5]Core Facility Center, CAS Center for Excellence in Molecular Plant Sciences, Chinese Academy of Sciences, Shanghai, China. [6]CAS Key Laboratory of Quantitative Engineering Biology, Shenzhen Institute of Synthetic Biology, Shenzhen Institutes of Advanced Technology, Chinese Academy of Sciences, Shenzhen, China. [7]Key Laboratory of Systems Health Science of Zhejiang Province, School of Life Science, Hangzhou Institute for Advanced Study, University of Chinese Academy of Sciences, Hangzhou, China. [8]These authors contributed equally: Xinyun Jing, Niubing Zhang, Xiaojuan Zhou, Ping Chen, Jie Gong. ✉e-mail: lixuan@sippe.ac.cn

occupancy at each position, and the spacing of nucleosomal arrays often fluctuate around actively transcribed genes, modulating the binding of transcription factors and the passing through of RNA polymerases[3,12–14].

Eukaryotic histones (components of the eukaryotic nucleosome) are believed to have originated and evolved from archaeal families of DNA-binding proteins, also called archaeal histone-like proteins or archaeal histones[15–17]. Many archaeal histone-like proteins have been resolved by X-ray crystallography, and some were shown to have the same structural fold as eukaryotic histones[17,18]. However, the higher-order structure of the eukaryotic nucleosome, i.e., the octasome–DNA complex, is unique to eukaryotic cells and has not been found in organisms outside the eukaryotic domain. Although nucleosomes play a central role in packaging genomic DNA and regulating genome accessibility in eukaryotic cells, the origin and evolution of the higher-order nucleosome complex are not well understood. In particular, how archaeal and bacterial ancestors of eukaryotes would respond to the evolutionary transition from archaeal histones toward nucleosome complexes remains a mystery. We are intrigued by the question if we could 're-create' the higher-order nucleosome complex in prokaryotic organisms, to investigate the transition process. This synthetic biology approach, 'building from scratch,' would establish a platform to study the assembly of the nucleosome complex in a 'living' prokaryotic system, helping researchers gain insight into the evolutionary origin of the eukaryotic nucleosome.

This approach is potentially rewarding but also potentially challenging, as many obstacles remain. First, the formation of eukaryotic nucleosomes in bacteria might render the host nonviable, as the nucleosome complex might interfere with the way the bacterial genome is organized and configured, as well as how gene activities are regulated. The bacterial cell division cycle might be severely perturbed in the presence of eukaryotic nucleosomes. However, we were encouraged by the earlier work of Warnecke and colleagues, who successfully expressed the archaeal histone-like protein, i.e., HMfA or HMfB, in *Escherichia coli*, which had only a mild impact on cell growth in short-term culture experiments[19]. Another concern is that in vivo assembly of eukaryotic nucleosome complexes has not been successfully attempted in bacteria, likely owing to the distinct cellular environments of bacteria. We were inspired by the work of Shim and coworkers, who coexpressed the *Xenopus* core histones H2A, H2B, H3, and H4 in bacteria for purification[20]. In this study, we successfully engineered the in vivo assembly of the nucleosome core in the model bacterium *E. coli* and showed that the host chromosomal DNA and eukaryotic histones can assemble to form nucleosome complexes with many features resembling those found in eukaryotic cells. Furthermore, we demonstrated that the nucleosome-forming bacterium can accommodate a measurable level of eukaryotic nucleosomes in their genome and have sustained cell division in long-term growth experiments under specified conditions.

## Results

### In vivo assembly of the eukaryotic nucleosome core in *E. coli*
To enable the assembly of eukaryotic nucleosomes in *E. coli*, we generated polycistronic constructs for the coexpression of four core histones, H2A, H2B, H3, and H4, using *Xenopus* genes (Fig. 1a and Supplementary Fig. 1a). The sequences of the *Xenopus* core histones were codon-optimized for *E. coli* expression. Their expression construct, pET-Xen, was tested using two *E. coli* strains, BL21(DE3) (Supplementary Fig. 1b) and Rosetta(DE3) (Fig. 1b). The *Xenopus* histones were soluble and were found mainly in supernatants when their expression was induced by isopropyl β-D-thiogalactopyranoside (IPTG; 400 μM), indicating that the eukaryotic histones folded correctly in the *E. coli* cytoplasm. We also tested different culture temperatures, 18 °C and 37 °C, induced by 400 μM IPTG, and found that *Xenopus* histones remained consistently soluble at both temperatures (Supplementary Fig. 1c).

We next tested whether eukaryotic core histones can assemble and form nucleosome complexes in vivo in *E. coli* cells given the very different cellular environment of the bacterium. A micrococcal nuclease (MNase) digestion assay is typically used to detect nucleosome formation and positioning in eukaryotic cells[21,22]. To examine nucleosome formation in bacteria, we developed an in situ micrococcal nuclease digestion assay for use with *E. coli* cells (ecMNase assay, Fig. 1c). We applied the ecMNase assay to the *Xenopus* histone-expressing strain Ec-r-pXen and the control strain Ec-r-pET29a containing empty plasmid pET29a, which were treated with 400 μM IPTG. To our excitement, we detected a unique DNA fragmentation pattern due to the protection of *E. coli* genomic DNA by nucleosomes, similar to the fragmentation pattern of eukaryotic genomic DNA in the MNase assay (Fig. 1d). This pattern was absent in the control strain Ec-r-pET29a. These results provide one piece of strong evidence that eukaryotic histones can assemble and form nucleosome complexes in bacterial cells. Note at a 37 °C growth temperature, which enabled faster *E. coli* growth, the assembly of nucleosome complexes was more efficient than at 18 °C (Supplementary Fig. 1d). In our micrococcal nuclease assay of strain Ec-r-pXen grown at 37 °C, we observed nuclease-resistant DNA bands corresponding to mono-, di- and tri-nucleosomes. In cells grown at 18 °C, we saw only nuclease-resistant DNA bands corresponding to mono-nucleosomes (Supplementary Fig. 1d). Subsequently, we used the *Xenopus* histone expressing *E. coli* strain Ec-r-pXen grown at 37 °C for all experiments.

The mono-nucleosome band of Ec-r-pXen had a size of ~40 to 150 bp (a major peak was confirmed to be ~146 bp by ecMNase-seq; see next section) (Fig. 1d), which is approximately the same size of DNA observed in micrococcal nuclease assays of *Xenopus* DNA[23]. The size of di-nucleosome protected DNA, close to 300 bp, appeared smaller than that for *Xenopus* ones. To demonstrate the formation of nucleosome complexes in *E. coli* cells, we used an approach similar to that previously used for the isolation of native chromatin fragments from yeast[24]. Nucleosomes were purified via two chromatography steps, under non-denaturing conditions ("*Methods*"). Nucleosomes of different compositions were separated using a Superdex 200 Increase 10/300 GL column (Cytiva Life Sciences) (Fig. 1e, top). Peak fractions were further analyzed with sodium dodecyl sulfate-polyacrylamide gel electrophoresis (SDS-PAGE), revealing the presence of the four nucleosome components, H2A, H2B, H3, and H4 in fractions of 27–42 (Fig. 1e, bottom).

To prove that the protected fragmentation of *E. coli* genomic DNA is dependent on the formation of nucleosomes, we generated two constructs in which histone H3 was either deleted (pET-Xen-H3del) or truncated by removing amino acid residues 101–132 (pET-Xen-H3Δ) (Fig. 1f). While the remaining histones, H2A, H2B, and H4, were normally expressed (Fig. 1g), both the H3 deletion construct (Ec-r-pXen-H3del) and the H3 truncation construct (Ec-r-pXen-H3Δ) failed to form nucleosome-DNA complexes and resulted in the complete absence of protected fragments of *E. coli* genomic DNA (Fig. 1h).

We further visualized the nucleosome core complexes on *E. coli* genomic DNA using atomic force microscopy (AFM). The protoplasts were prepared from *E. coli* cells and then lysed with ecMNase buffer to release DNA-nucleosome complexes for AFM analysis as described (*Methods*). For the eukaryotic nucleosome-forming strain (Ec-r-pXen), we observed beads-on-a-string like structures along DNA strands. They represent the nucleosome core array, which is similarly observed in the positive control from *Xenopus* liver cells (Fig. 1i and Supplementary Fig. 1e, f). In contrast, no nucleosome complexes were detected in the genomic DNA of samples from the control *E. coli* strain (Ec-r-pET29a). Taken together, these data demonstrate the successful in vivo assembly of eukaryotic nucleosome core complexes in *E. coli*.

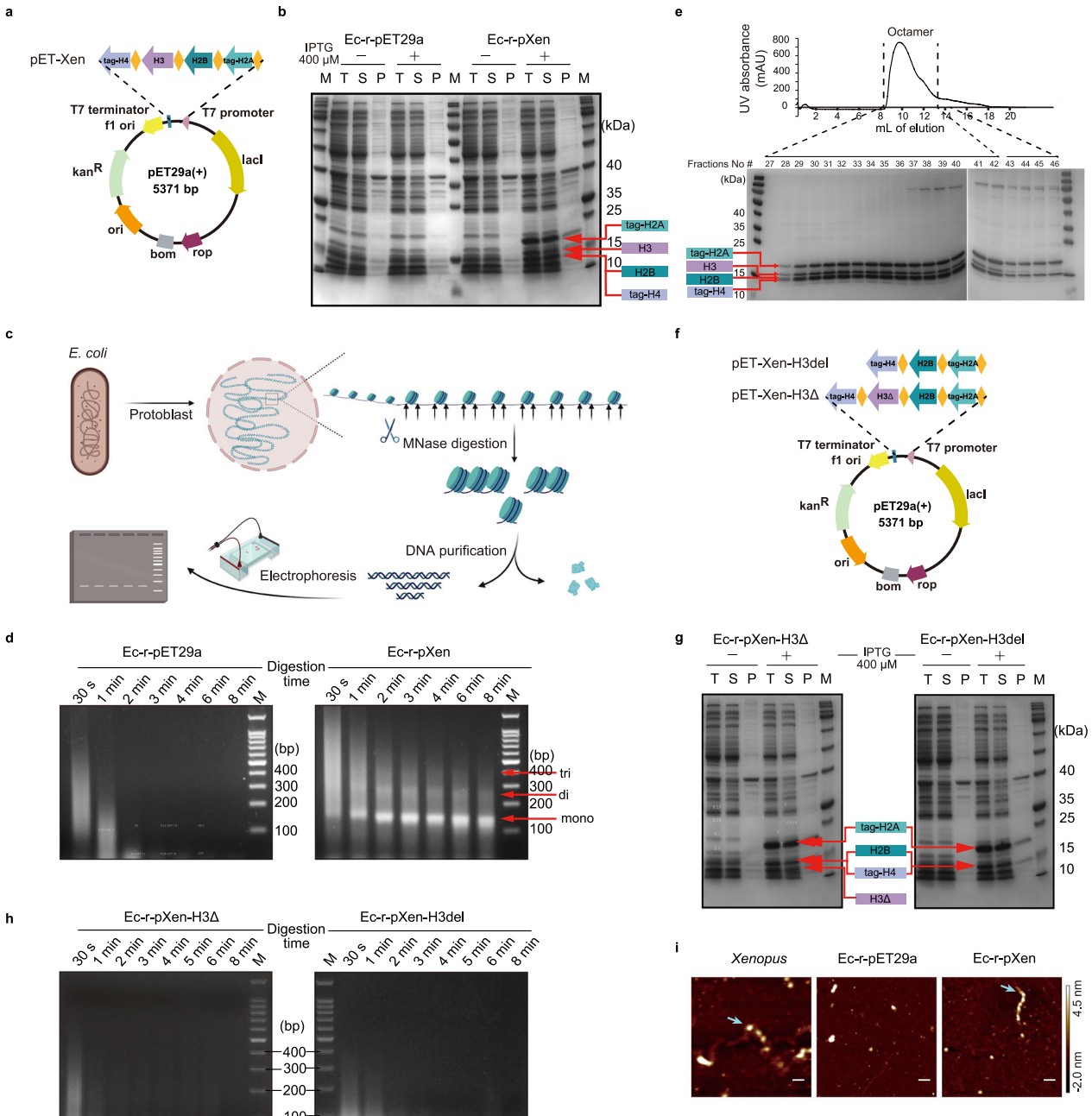

**Fig. 1 | In vivo assembly and detection of eukaryotic nucleosome core in *E. coli*.** A schematic of the polycistronic construct for expression of *Xenopus* histones in *E. coli*. In (**a**, **b**, **e**, and **f**), "tag-H2A" is a histone H2A construct with an N-terminal S-tag and a hexahistidine (6 × His) tag, while "tag-H4" is a histone H4 construct with a C-terminal 6 × His tag. **b** Expression of *Xenopus* histones in *E. coli* Rosetta(DE3) grown at 37 °C, induced with (+) or without (−) 400 µM IPTG. T, total lysate; S, supernatant; P, pellet; M, protein standards. tag-H2A, 18.15 kDa; H2B, 13.64 kDa; H3, 15.07 kDa; tag-H4, 13.31 kDa. The figure is representative of three independent experiments. **c** A schematic of in situ micrococcal nuclease digestion assay for *E. coli* cells (ecMNase assay). **d** DNA fragmentation profiles of two *E. coli* strains grown at 37 °C, induced with 400 µM IPTG, using ecMNase assay. Arrows mark mono-, di- and tri-nucleosome bands. M, DNA standards. In (**b**) and (**d**), Ec-r-pET29a represents the control strain with pET29a, while Ec-r-pXen represents the nucleosome-forming strain. The figure is representative of three independent experiments. **e** Elution diagram of the nucleosome complexes from the size exclusion

chromatography monitored by UV (280 nM) absorption (top), and SDS-PAGE analysis of fractions 27–46 (bottom). The 8–15 mL elution peak corresponds to these fractions. The figure is representative of three independent experiments. **f** A schematic of two polycistronic constructs for expression of *Xenopus* histones with H3 either deleted (pET-Xen-H3del) or truncated between amino acid residues 101–132 (pET-Xen-H3Δ). **g** Expression of *Xenopus* histones in *E. coli* strains Ec-r-pXen-H3del and Ec-r-pXen-H3Δ containing pET-Xen-H3del and pET-Xen-H3Δ, grown at 37 °C, induced with (+) or without (−) 400 µM IPTG, respectively. H3Δ, 11.55 kDa. The figure is representative of three independent experiments. **h** DNA fragmentation profiles of strains Ec-r-pXen-H3del and Ec-r-pXen-H3Δ using ecMNase assay. **i** AFM images of nucleosome complexes (bead-like structure) in *Xenopus* (left) and Ec-r-pXen (right), but not in Ec-r-pET29a (middle). The height profiles were provided in Supplementary Fig. 1e. Arrows indicate the nucleosome core. Bar length = 100 nm. The figure is representative of two independent experiments. Source data are provided as a Source Data file.

### In vivo histone octamer assembly visualized using split green fluorescent protein (splitGFP)

The tripartite split green fluorescent protein (splitGFP) system was previously developed to detect the formation of protein complexes in vivo and to resolve their subcellular localization[25]. The tripartite splitGFP system can be calibrated to detect transient protein-protein interactions and low-affinity complexes in *E. coli*[26,27]. We sought to visualize the in vivo assembly of histone octamer complexes in *E. coli* using the tripartite splitGFP system, as depicted in Fig. 2a. Previously, it was shown that the tripartite splitGFP system did not perturb the expression of associated proteins in *E. coli* and enabled the analysis of protein associations in vivo[26,28]. To make the histone octamer complexes in *E. coli* fluoresce, we tethered the splitGFP fragments GFP1-9, GFP10, and GFP11 to H2A, H2B, and H4, respectively. Considering the topology of octasomes based on the model structure of the eukaryotic nucleosome, we placed GFP1-9, GFP11, and GFP10 at the N-termini of H2A [GFP(1–9)-H2A] and H4 (GFP11-H4), and at the C-terminus of H2B (H2B-GFP10), respectively (Fig. 2b).

The inclusion of splitGFP to *Xenopus* histones did not affect the formation of histone octamer complexes, as evidenced by the ecM-Nase results from the Ec-r-pXen-spGFP strain containing the pET-Xen-spGFP vector, which expressed GFP(1–9)-H2A, H2B-GFP10, H3, and GFP11-H4 upon 400 μM IPTG induction (Fig. 2c). While the mono- and di-nucleosome bands were visible, the tri-nucleosome was less visible compared to that in Fig. 1d. A likely reason is that the splitGFP caused the octamers became less stable. As expected, in the Ec-r-pXen-spGFP-H3Δ strain containing the pET-Xen-spGFP-H3Δ vector, which encodes a truncated H3, no nucleosome-protected DNA fragments were observed.

To visualize the in vivo assembly of histone octamer complexes in *E. coli*, the *E. coli* cells of the different strains were grown to log phase and treated with 400 μM IPTG for 60 min before they were stained with DAPI and observed using a Nikon AX R confocal microscope. Using a predetermined setting, the Ec-r-pXen-spGFP cells showed green fluorescence upon IPTG induction (Fig. 2d, second-row panels). The control Ec-r-pET29a cells did not fluoresce under the same conditions. Importantly, the green fluorescence signal overlapped with the DAPI signal, indicating that the fluorescent nucleosomes co-localized with the genomic DNA in the Ec-r-pXen-spGFP cells. Furthermore, the fluorescence was dependent on histone octamer formation, as the Ec-r-pXen-spGFP-H3Δ cells with dysfunctional H3 did not fluoresce under IPTG induction. Given all the evidence from ecMNase assay, gel filtration, atomic force microscopy, and the use of splitGFP constructs, these results corroborate the successful in vivo assembly of eukaryotic nucleosome core in a bacterial system.

### Titration of nucleosome formation in *E. coli*

The in vivo nucleosome assembly in Ec-r-pXen was titrated by varying the histone expression levels with IPTG induction concentration in the range of 1 to 400 μM, followed by prolonged micrococcal nuclease digestion to determine how much *E. coli* DNA was protected from digestion by being part of nucleosomes (Fig. 3a). With increasing IPTG concentration, the amounts of histones and nucleosome-protected DNA both increased, indicating a positive correlation between histone synthesis and nucleosome formation in *E. coli* cells (Fig. 3b–d).

To investigate the organization of nucleosomes assembled in *E. coli*, we carried out ecMNase-seq analysis on IPTG-titrated Ec-r-pXen cells. DNA fragments around the mono-nucleosome length were extracted for Illumina sequencing (Fig. 3c). The ecMNase-seq reads contained two populations separated by a valley at ~140 bp (Supplementary Fig. 2). The larger population had a major peak at 146–147 bp. In addition, a minor peak at ~154 bp was apparent, which was also featured in MNase experiments for yeast and mouse liver chromatin lacking H1 histone[29]. Notably, the sub-nucleosomal population (<140 bp) sometimes displayed minor bumps at 105–106, 115–116,

125–126, or 135–136 bp. (Notably, these features became more apparent in the long-term growth experiment.) The 10-bp periodicity between the minor peaks observed within the sub-nucleosomal population is a typical structural feature commonly associated with eukaryotic nucleosomes, representing products from internal digestion of nucleosomal DNA[29,30].

Notably, 115–116 bp fragments have been proposed to be associated with hexasomes in eukaryotes[31,32]. Hexasomes were previously shown to lack one of the two H2A/H2B from octasomes and to be wrapped in DNA approximately 30 bp shorter than that of octasomes[31,33]. Hexasomes are believed to be a dynamic transient structure in nucleosome assembly or disassembly[34,35]. The proportion of the sub-nucleosomal population decreased with increasing IPTG concentration in the titration experiments (Supplementary Fig. 2). Increasing histone levels shifted the equilibrium of nucleosome assembly toward intact states that were less accessible for internal digestion by micrococcal nuclease.

We next mapped the ecMNase-seq reads from the IPTG-titration experiments to the *E. coli* genome using the sequence quantity for each IPTG concentration in proportion to the titration results (Fig. 3d and Supplementary Data 1). The distribution of nucleosome-protected DNA fragments on the *E. coli* genome was ubiquitous, containing regions with divergent mapping coverage (Fig. 3e). Nucleosome peaks were called using the peak calling program Danpos[36]. We found that the number of nucleosome peaks remained similar (~22,000) for various IPTG concentrations, despite the different quantities of reads used in peak calling (Supplementary Data 1). The peak densities for different IPTG concentrations showed similar variations (Fig. 3f left), and their distributions were highly correlated ($r \geq 0.92$) across the *E. coli* genome (Fig. 3f right). The nucleosome peak positions from experiments with different IPTG concentrations were highly coincident (Fig. 3g). Although eukaryotic nucleosomes are alien structures to *E. coli*, the genome-wide distribution data suggest that nucleosome positioning is not random but largely consistent among bacteria with different histone expression levels.

### Long-term growth of nucleosome-forming *E. coli*

The success of in vivo assembly of the nucleosome core in *E. coli* prompted us to test whether the nucleosome-forming bacterium can divide and sustain growth. In preliminary experiments, we found that the *E. coli* strain newly transformed with the plasmid pET-Xen could divide to form colonies the next day after overnight growth when plated on LB plates supplemented with IPTG (1 μM) as well as on no-IPTG control plates. However, they failed to form colonies on plates with elevated IPTG concentrations, i.e., 10 μM and up. We repeated the experiment and randomly picked six colonies (designated as Ec-r-pXen clones) from the 1 μM-IPTG plate to grow them in liquid medium supplemented with IPTG (1 μM) and kanamycin (50 μg/mL) before performing the ecMNase assay. To our delight, all six clones (6/6) were confirmed to be positive for nucleosome formation (Supplementary Fig. 3a). This indicates that *E. coli* with eukaryotic nucleosomes is viable and can divide and grow under the tested conditions.

We then initiated a longer-term growth experiment using four of the six nucleosome-forming clones with two Ec-r-pET29a clones as control, which were passaged continuously in a liquid culture medium containing IPTG (1 μM) and kanamycin (50 μg/mL) (Fig. 4a). The cultures were examined by an ecMNase assay at passages 2, 6, 10, and 14. All four Ec-r-pXen clones maintained nucleosomes throughout the course of the experiment (Supplementary Fig. 3b). We estimated that by passing 14 times, the nucleosome-forming *E. coli* underwent more than 110 divisions based on the measured optical density changes between the start and end of each passage. To characterize the nucleosome-forming *E. coli* in long-term growth, we performed ecMNase-seq, mRNA-seq, and whole-genome sequencing (WGS) on cultures from passages 2, 6, 10, and 14.

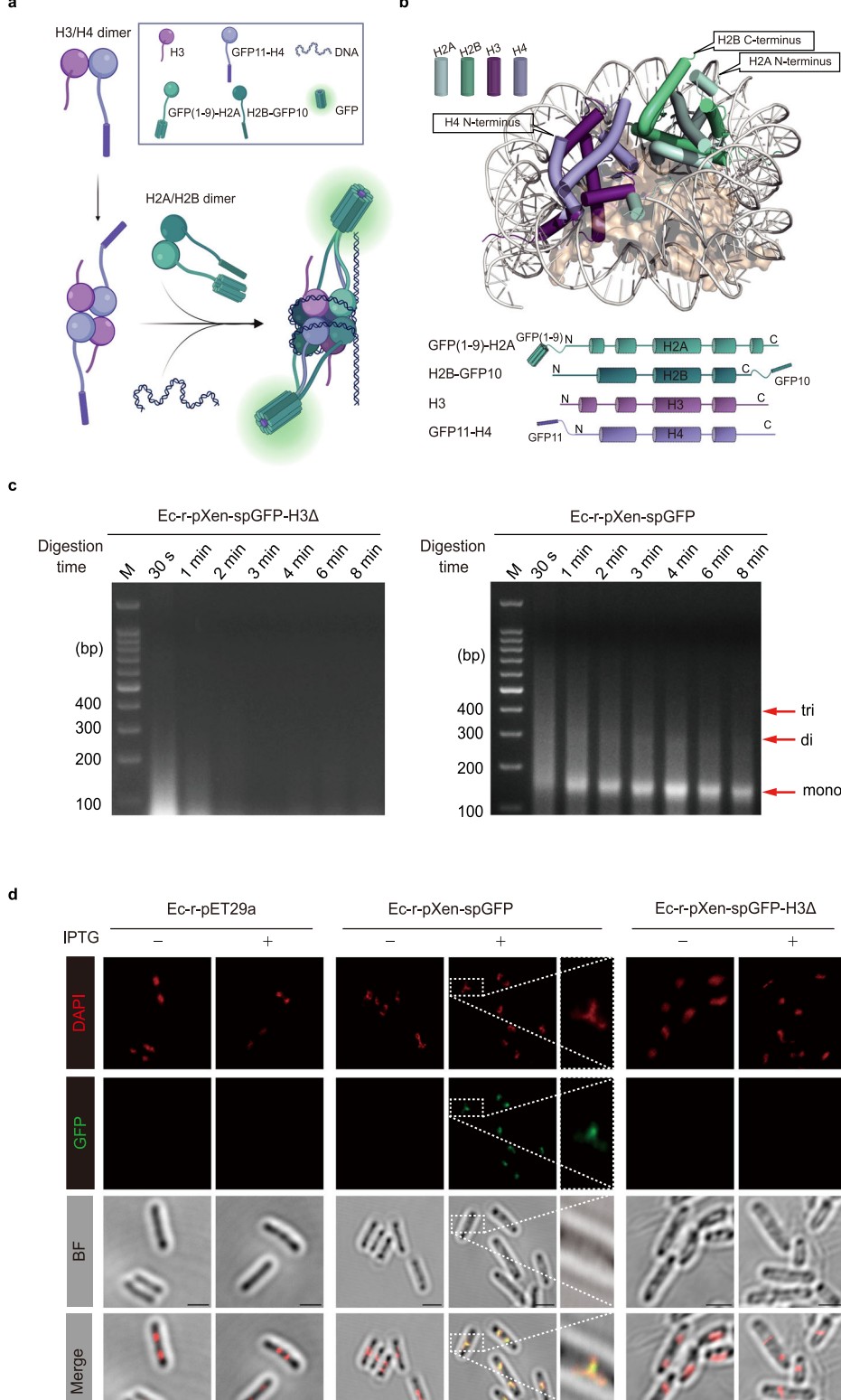

**Fig. 2 | Visualization of in vivo nucleosome formation in *E. coli* using tripartite split GFP (spGFP). a** Schematic representation of the spGFP system for detection of in vivo assembly of the octasome in *E. coli*. **b** The topology of *Xenopus* octasome structure (PDB:3MGQ) (top) and design of spGFP fragment-*Xenopus* histone fusion proteins (bottom). spGFP fragments are marked as GFP(1–9), GFP10, and GFP11 that links to the N-terminus of H2A, C-terminus of H2B, and N-terminus of H4, respectively. **c** DNA fragmentation profiles of strains Ec-r-pXen-spGFP-H3Δ and Ec-r-pXen-spGFP, grown at 37 °C, induced with 400 μM IPTG, using ecMNase assay. The

complete gel pieces are provided in Supplementary Fig. 5c. The figure is representative of three independent experiments. **d** Light microscopy fluorescence images of *E. coli* strains Ec-r-pET29a, Ec-r-pXen-spGFP and Ec-r-pXen-spGFP-H3Δ treated with 400 μM IPTG (+) or without IPTG (−) ("*Methods*"). DAPI was used to stain *E. coli* DNA. BF, bright field. Bar length = 2 μm. The rightmost field for Ec-r-pXen-spGFP is the magnified dotted square area in the middle field. The figure is representative of three independent experiments. Source data are provided as a Source Data file.

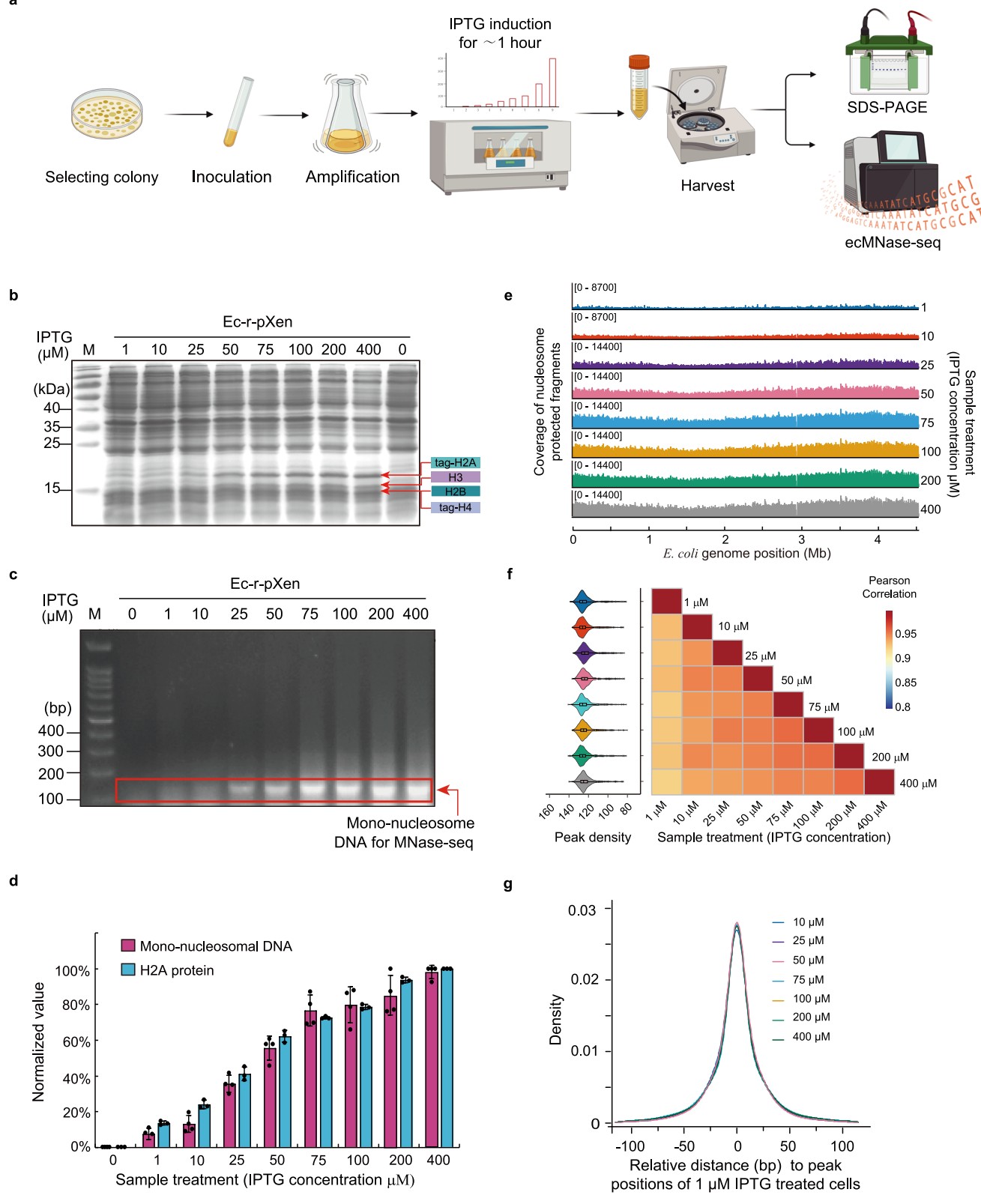

To characterize the nucleosome organization in *E. coli* during long-term growth, we analyzed the mRNA-seq data, and found that the expression of the core histone genes remained stable over the course of the 14 passages (Fig. 4b right axis). Notably, the abundance of histones' poly-cistron transcripts was ranked on average 53rd out of 4160 gene transcripts in the long-term growth experiment (Supplementary Data 2). In comparison, several *E. coli* proteins (i.e., RNA polymerase

alpha subunit, DNA polymerase subunits: alpha, beta, and epsilon, and DNA-binding protein HU subunits: alpha and beta) that are involved in transcription and DNA replication process, were ranked 15th, 1024th, 374th, 1333rd, 1378th and 3106th, respectively. Furthermore, the ecMNase-seq data from all four Ec-r-pXen clones displayed consistent profiles along passages 2, 6, 10, and 14 (Supplementary Fig. 4). Nucleosome peaks were called and found to have numbers in a range

**Fig. 3 | In vivo nucleosome assembly with varying histone expression levels in _E. coli_. a** A schematic of experimental design for titration of in vivo nucleosome assembly in _E. coli_ by varying _Xenopus_ histones expression levels through inducing with different IPTG concentrations for ~ 1 h. **b** SDS-PAGE analysis for expression of _Xenopus_ histones, induced by IPTG at different concentrations for ~ 1 h. **c** ecMNase assay for titration of nucleosome formation at different IPTG concentrations, for which prolonged MNase digestion conditions were used. M, DNA standards. **d** Quantification of histones expression levels represented by tag-H2A (**b**) and mono-nucleosome DNA fragments (**c**) in _E. coli_ at different IPTG concentrations. Data are normalized with the maximum value as 100%, and represent mean ± SEM of three independent experiments (_n_ = 3) for H2A protein, and four independent experiments (_n_ = 4) for mono-nucleosomal DNA. **e** Genome-wide coverage of nucleosome-protected fragments along the _E. coli_ genome in IPTG-titration experiments. Peak height represents the coverage depth of nucleosome-protected

fragments. Values in brackets represent the range of the _Y_-axis. The identity of each sample is listed to the right. **f** (left) Violin plots with boxplots showing the peak density variation in Ec-r-pXen cells at different IPTG concentrations, and (right) correlation matrix of peak density distribution between paired IPTG concentrations. Peaks were called for each IPTG concentration using reads randomly drawn from two biological replicates (_n_ = 2) (Methods). The violin plots show the summary statistics (the median, first and third quartiles, 1.5× interquartile range) and kernel distribution of peak density (_n_ = 912). **g** Density plot for nucleosome peaks relative to those of 1 μM IPTG treated cells. Nucleosome peaks from Ec-r-pXen cells treated with 10, 25, 50, 75, 100, 200, and 400 μM of IPTG are aligned to the center of closest peaks from cells treated with 1 μM of IPTG. For panels (**e**–**g**), because no sufficient mono-nucleosome protected DNA from 0 μM IPTG treatment was recovered for MNase-sequencing under the same conditions, no 0 μM IPTG treatment data from Ec-r-pXen were presented. Source data are provided as a Source Data file.

similar to those from the 1 μM IPTG induction experiment (Fig. 4b left axis and Supplementary Data 3). Notably, the nucleosome peak numbers remained stable throughout all passages. Compared to that in the 1 μM IPTG-induction experiments, the sub-nucleosomal population (< 140 bp), in the long-term growth data was reduced, in which the minor peaks at 105–106, 115–116, 125–126, and 135–136 bp became apparent (Supplementary Fig. 4).

We plotted the nucleosome occupancy for dyads in the same order based on their occupancy ranking from high to low at passage 14. The nucleosome occupancy for dyads was remarkably consistent along passages 2, 6, 10, and 14 (Fig. 4c, d). The high correlation of nucleosome occupancies on corresponding dyads between passages provided evidence of stable nucleosome formation in the long-term growth experiment. The GC content plot for the dyads indicated a certain correlation between nucleosome occupancy and GC content (Fig. 4c, right panel).

Nucleosomes are known to form unique patterns within eukaryotic gene structures[3]. To reveal how eukaryotic nucleosomes assemble in genic regions in _E. coli_, we plotted the average nucleosome frequency flanking the transcription start site (TSS) for all _E. coli_ genes. The nucleosome footprints along the _E. coli_ genes resembled those of typical eukaryotic genes in yeasts, humans, etc[37,38], for which a nucleosome-free region (NFR) near the TSS is flanked by upstream and downstream nucleosomes, with distal nucleosomes becoming less phased in positioning (Fig. 4e). Notably, the nucleosome profile around the TSS remained generally unchanged along the course of long-term culture. Examples of some genic regions, such as cynR, _armB_, _cdaR_, and _pstS_, showed different phasing profiles in the nucleosome-forming _E. coli_, while all of them remained consistent throughout the passages (Fig. 4f).

To evaluate the stability of global gene expression in the nucleosome-forming _E. coli_ during long-term growth, we conducted transcriptome analysis using mRNA-seq data across the different passages. The transcriptome of each _E. coli_ clone remained highly consistent across passages (Ec-r-pXen clone #1; Fig. 4g), and the Pearson correlations between different passages were greater than 0.97, 0.95, 0.92, and 0.93 for clones #1, #2, #3 and #4, respectively. Compared with the control strain (Ec-r-pET29a) treated with IPTG (1 μM), the nucleosome-forming (Ec-r-pXen) strain. We found 70 DEGs in the Ec-r-pXen strains that were common to all passages, comprising 37 up-regulated and 33 down-regulated genes (Supplementary Data 4). The down-regulated genes primarily fell into categories such as amino acid metabolism, glycerol degradation, pyrimidine degradation and purine degradation, D-glucarate degradation, and glucose degradation (Supplementary Data 4). Among the up-regulated genes, in addition to _Xenopus_ histone genes, 15 genes were associated with _Escherichia_ phage DE3, five with heat shock and chaperone proteins, one with a small RNA gene regulating oxidative and osmotic stress responses, four with ferric enterobactin, β-glucoside, ferric dicitrate, and inorganic phosphate transport, and one with sulfate activation for

sulfonation and assimilatory sulfate reduction, and one gene encoding an allantoinase enzyme. The DEGs included many stress response genes in _E. coli_[39–41], indicating that nucleosome formation in _E. coli_ imposes some burden on host cells.

We further investigated whether genetic mutations accumulated in the Ec-r-pXen clones during long-term growth using whole-genome sequencing (WGS). We analyzed the WGS data from different passages for clones #1, 2, 3, and 4. Compared to the control strain (Ec-r-pET29a), only Ec-r-pXen clone #4 had two SNPs that were found in at least two passages with an allele frequency (AF) ≥10% but were not present in the control (Supplementary Data 5). However, we failed to validate these SNPs using PCR amplification and Sanger sequencing. Therefore, in all likelihood, they represent artifacts generated from Illumina sequencing rather than genuine mutations that arose during long-term growth.

### Estimate of nucleosome level and morphology and growth fitness of nucleosome-forming _E. coli_

We also estimated the histone protein abundance in the same nucleosome-forming cells using liquid chromatography-tandem mass spectrometry by multiple reactions monitoring (LC-MRM-MS) analysis on its proteome. The unique peptides for histones H2A, H2B, H3 and H4 were identified for Ec-r-pXen (Supplementary Data 6), but not found in the control Ec-r-pET29a (Supplementary Data 7). The protein levels of histones H2A, H2B, H3, and H4 in the strain Ec-r-pXen were estimated using emPAI (Exponentially Modified Protein Abundance Index)[42], which were ranked 542nd, 578th, 202nd, and 144th among 1493 proteins identified, respectively.

The effect of nucleosome formation on cell morphology and growth fitness was evaluated by comparing the nucleosome-forming strain (Ec-r-pXen) from passage 2 and the wild-type control strain (Ec-r-pET29a). The Ec-r-pXen cells displayed a normal cell shape but had a slight increase in outliers with longer cell lengths (Fig. 5a, b). To our surprise, the growth rate of the Ec-r-pXen cells in rich medium was comparable to that of the control (Fig. 5c). However, the growth competition assay revealed a rapid loss of Ec-r-pXen cells in co-culture with the control (Ec-r-pET29a) (Fig. 5d). After the start of co-culture, the percentage of Ec-r-pXen cells decreased from 50% to ~ 15% by 72 h and to ~ 4% by 120 h. Therefore, nucleosome-forming _E. coli_ appears to have lower fitness in growth competition with wild-type _E. coli_.

### Phenotypic analysis of nucleosome-forming _E. coli_ under various conditions

The growth phenotype of the nucleosome-forming strain (Ec-r-pXen) from passage 2 was evaluated in rich medium (LB), low-nutrient medium (R2A), and variants of the complete synthetic medium M9 (M9, M9 with different carbon sources, and M9 with different nitrogen sources). The growth of Ec-r-pXen cells was slower than that of Ec-r-pET29a cells at all the tested temperatures, except for M9 with lactose as the carbon source (Fig. 6a). Interestingly, when lactose was used as

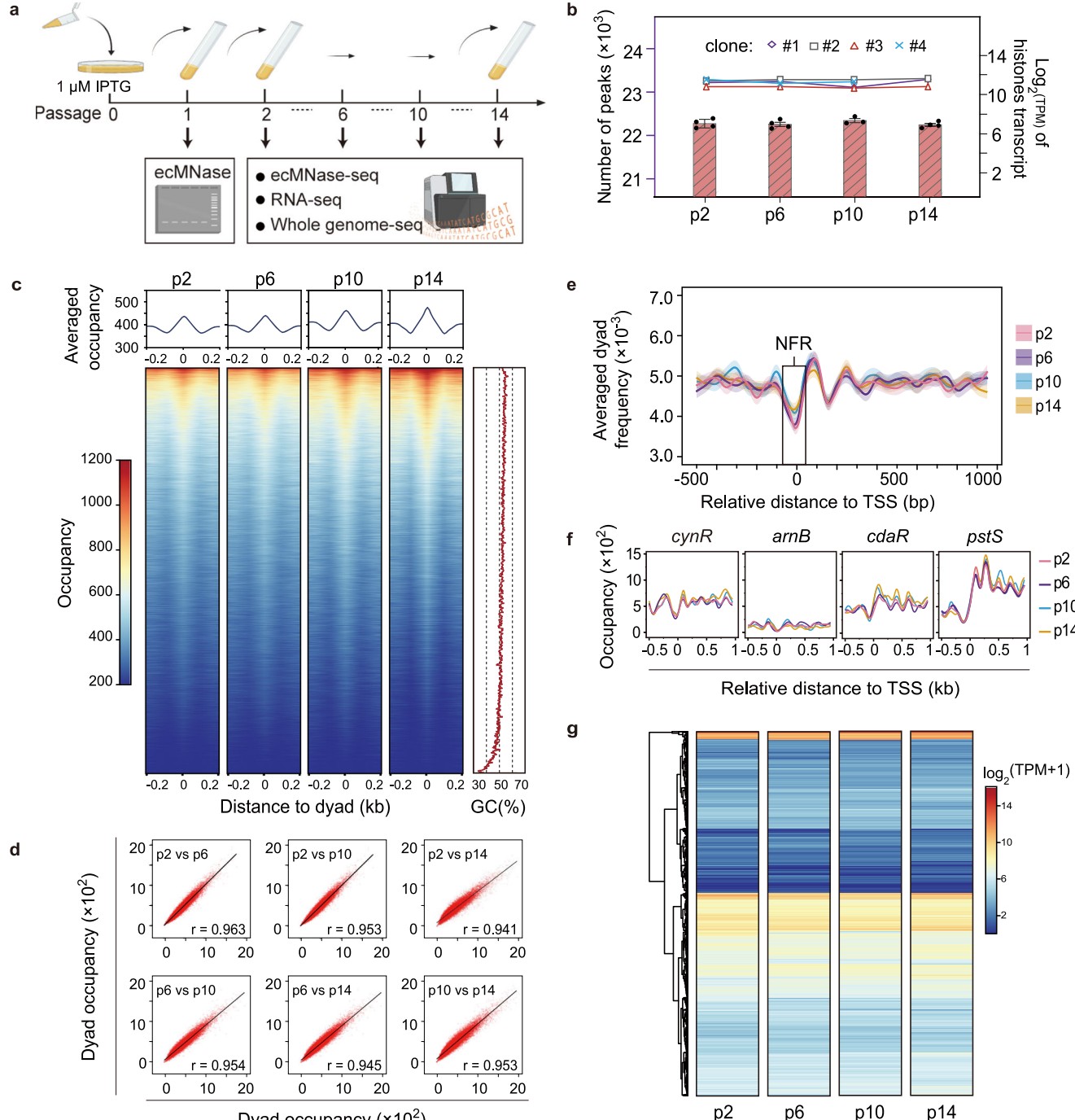

**Fig. 4 | Stable nucleosome formation in *E. coli* in long-term growth experiment.**
**a** A schematic for long-term passage and assay of the nucleosome-forming strain, Ec-r-pXen, grown at 37 °C, treated with 1 μM IPTG. **b** Histone gene expression (dot-line plot; Log₂TPM) and nucleosome peak number (bar plot; mean ± SEM) detected in the nucleosome-forming *E. coli* (clones 1–4) through the passages. p2, p6, p10, and p14 denote passages 2, 6, 10, and 14. TPM, transcript per million. Histone gene expression data: *n* = 4 biological replicates for p2, p6, and p10; *n* = 3 for p14. Nucleosome peak count data: *n* = 4 for p2, p6, and p14; *n* = 3 for p10. **c** Heatmaps and averaged profile (top panel) of dyad nucleosome occupancy for the nucleosome-forming *E. coli* from passages 2, 6, 10, and 14. The dyads from each passage are plotted in the same order based on their occupancy ranking in passage 14. The right panel shows the smoothed curve of the GC content for dyad sequences. For panels (**c**–**f**), data were generated from the *E. coli* clones 1–4 as those in panel (**b**). **d** Correlation of nucleosome occupancy between corresponding dyads for paired passages of the nucleosome-forming *E. coli*. r, Pearson's correlation

coefficient. **e** Averaged nucleosome distribution in genic regions of the nucleosome-forming *E. coli* from passages 2, 6, 10, and 14. Darker lines (smoothed) and associated lighter bands represent the mean and SEM of averaged dyad frequency, respectively. The distribution for each passage was derived from the data of 2448 transcripts. NFR, nucleosome-free region. **f** Nucleosome occupancy for genes *cynR*, *arnB*, *cdaR*, and *pstS*, from passages 2, 6, 10, and 14. For panels (**b**–**f**), for the long-term growth experiment, Ec-r-pET29a strains treated with 1 μM IPTG were used as control. Since no mono-nucleosome-protected DNA was recovered from them, no data for Ec-r-pET29a were included. **g** Heatmaps of the transcriptome (organized in hierarchical clustering) for the nucleosome-forming *E. coli* from passages 2, 6, 10, and 14 of clone 1, which is a representative result of four different clones. The color scheme represents log₂-transformed TPM. The Pearson correlations are greater than 0.97 for clone 1 among different passages (*n* = 4). Source data are provided as a Source Data file.

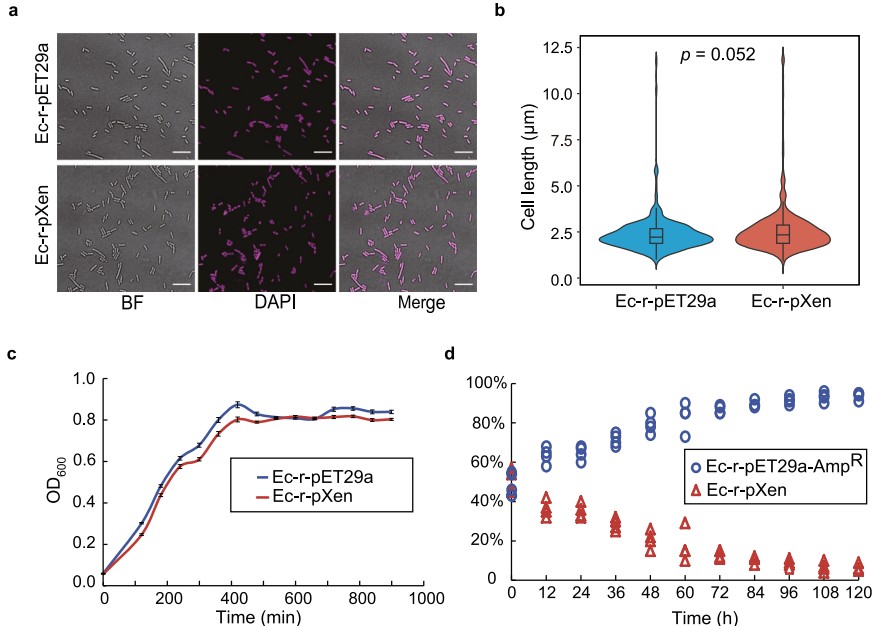

**Fig. 5 | Morphology and growth fitness of the nucleosome-forming *E. coli*.**
**a** Representative images for visualization of the nucleosome-forming (Ec-r-pXen) and control (Ec-r-pET29a) strains from an optical microscope ("*Methods*"). DAPI was used to stain *E. coli* DNA. BF, bright field. Bar length = 10 μm. *E. coli* cells were grown at 37 °C and treated with 1 μM IPTG for long-term growth. The figure is representative of two independent experiments. **b** Cell length distribution of the nucleosome-forming (Ec-r-pXen) and control (Ec-r-pET29a) strains. The violin plots show the summary statistics (the median, first and third quartiles, 1.5 × interquartile range) and kernel distribution of cell length. Plots are generated from the measurement of about 250 cells across 10 independent areas for each strain. A two-sided Mann-Whitney Rank Sum Test was performed to obtain the *p*-value. **c** Optical density measurement of cell growth (mean ± SEM) of the nucleosome-forming (Ec-r-pXen) and control (Ec-r-pET29a) strains at 37 °C in LB media with 1 μM IPTG. Data represent twenty-four biological replicates (*n* = 24). **d** Growth competition assay of the nucleosome-forming (Ec-r-pXen) and control (Ec-pET29a-amp^R) strains at 37 °C in LB media with 1 μM IPTG ("*Methods*"). Data represent four biological replicates (*n* = 4). Source data are provided as a Source Data file.

the carbon source, Ec-r-pXen grew slightly better than Ec-r-pET29a at 30 °C and 37 °C, but neither grew at 18 °C or 42 °C.

We further evaluated the growth fitness of nucleosome-forming *E. coli* at various temperatures combined with different pH or salt stress conditions (Fig. 6b, c). In rich medium (LB), the Ec-r-pXen cells had weaker growth fitness in general than the control (Ec-r-pET29a) under various combined conditions. These results indicated that nucleosome-forming *E. coli* exhibited a loss of fitness and adaptability to changing environmental conditions. However, compared with Ec-r-pET29a cells, Ec-r-pXen cells fared equally well toward certain chemical stressors, such as rifampicin (an RNA polymerase inhibitor), $H_2O_2$ (an oxidizing agent), and captothecin (a DNA-topoisomerase inhibitor), but had an erosion of growth fitness when treated with novobiocin (a DNA gyrase inhibitor) (Fig. 6d).

## Discussion

The eukaryotic nucleosome, a complex of eight histone proteins around which DNA is wrapped, plays a central role in organizing and compacting DNA and in regulating access to genomic information in eukaryotic cells. Despite its crucial importance, how the nucleosome complex emerged during eukaryogenesis and became a central component of eukaryotic cells, is not fully understood. In this study, we attempted to 're-create' a eukaryotic nucleosome in a living bacterium. By introducing synthetic genes encoding *Xenopus* histones, we engineered the in vivo assembly of the nucleosome core in *E. coli*. Under the condition that moderate histone expression was induced at 1 μM IPTG, we showed that nucleosome-forming *E. coli* is viable and has sustained growth for at least 110 divisions in long-term growth experiments under the specified conditions.

Our results showed that bacterial chromosome DNA and eukaryotic histones can assemble in vivo to form nucleosome core complexes with a degree of resemblance to those found in eukaryotic hosts. The formation of nucleosome complexes in *E. coli* has been corroborated by multiple lines of evidence, including data from ecMNase assays, the purification of nucleosome complexes, atomic force microscopy, and tripartite split green fluorescent protein experiments (Figs. 1, 2). Remarkably, the eukaryotic nucleosomes formed in *E. coli* cells had many features similar to those found in their native hosts. They had a ~146 bp mono-nucleosome length in the ecMNase assay, and a 10 bp periodicity for sub-nucleosomal products due to internal digestion of nucleosomal DNA. The nucleosome footprints in the *E. coli* genic regions also resembled those found in eukaryotic cells to a certain degree: for which an NFR region near the TSS is flanked by well-positioned nucleosomes at nearby and dissipated ones further away from the NFR (Fig. 4e). The similarity of the nucleosome arrays in *E. coli* genic regions to those in eukaryotic cells is intriguing. It is likely that the nucleosome profile around the genic regions of *E. coli* and of eukaryotic hosts may be shaped by some similar mechanisms, including the local sequence context, transcription factor binding, and transcription activities[43,44].

The eukaryotic nucleosome-forming *E. coli* are viable and exhibit sustained cell division in long-term growth experiments under the condition that histone expression was induced at 1 μM IPTG. Nucleosome formation at 1 μM IPTG was about 13.2% of that at 50 μM IPTG, and 7.4% of that at 400 μM IPTG (Fig. 3d). The moderate level of nucleosome formation in *E. coli* was corroborated by transcriptomic and proteomic abundance data (Supplementary Data 2, 6). The eukaryotic nucleosomes at the moderate level did not seem to disrupt or severely affect bacterial genome function in our long-term growth experiments. Furthermore, the stable nucleosome formation throughout the passages under the given conditions was corroborated with several lines of evidence, including *Xenopus* histone gene expression levels, nucleosome peak numbers, dyad position, and occupancy, nucleosome position and occupancy around transcribed

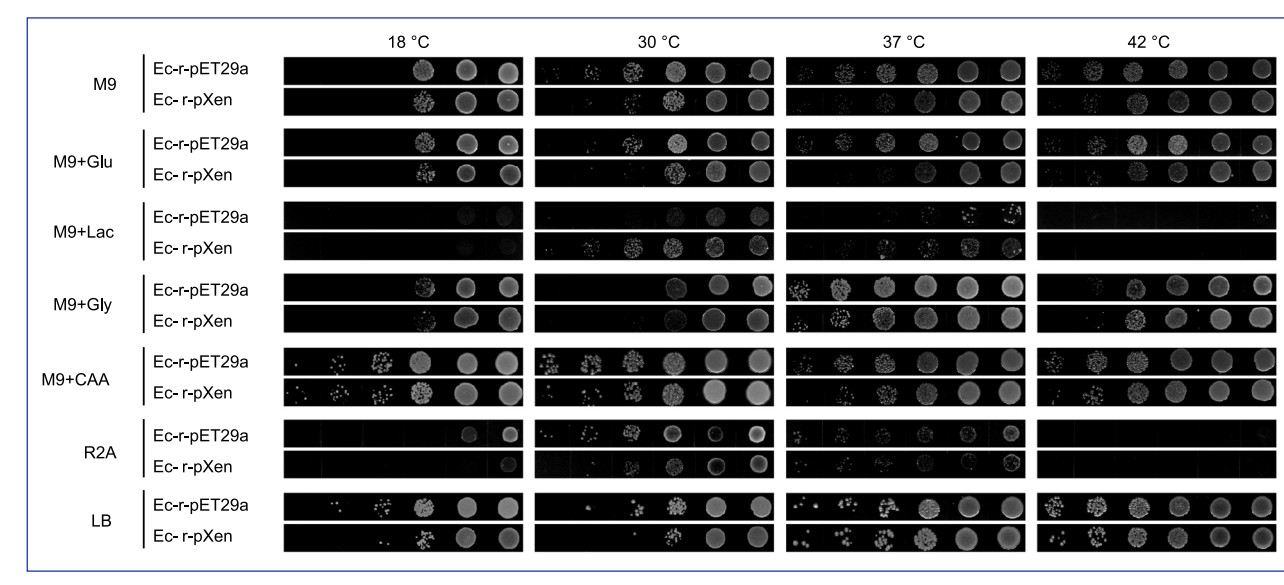

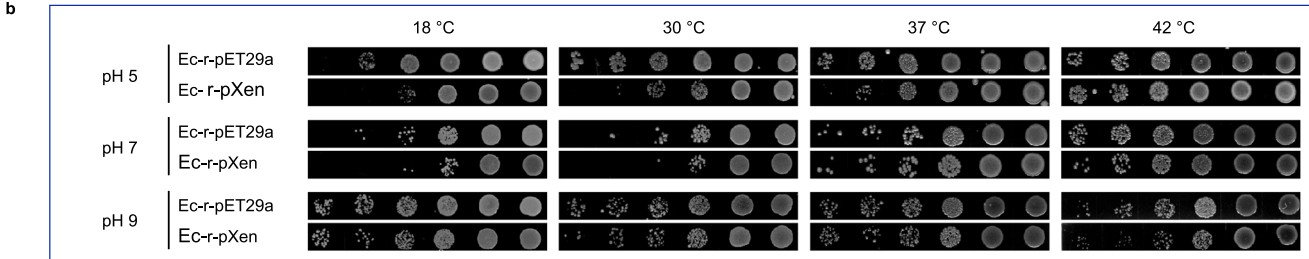

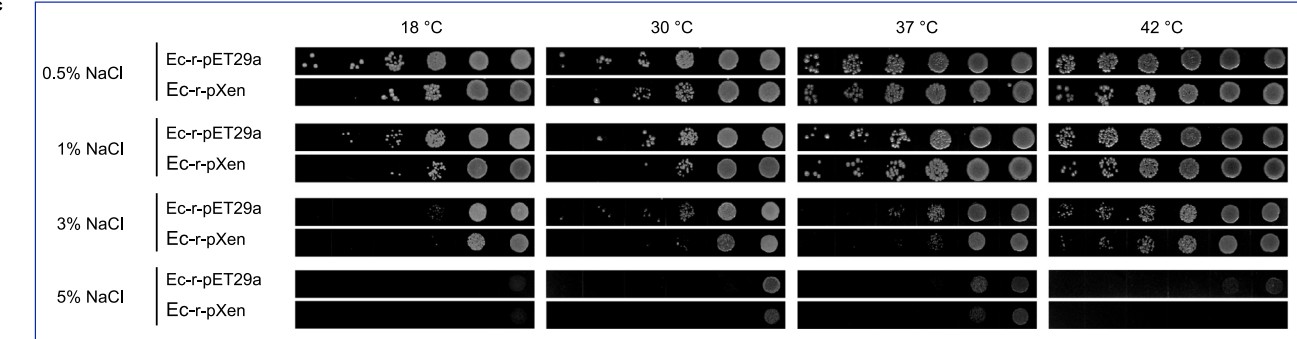

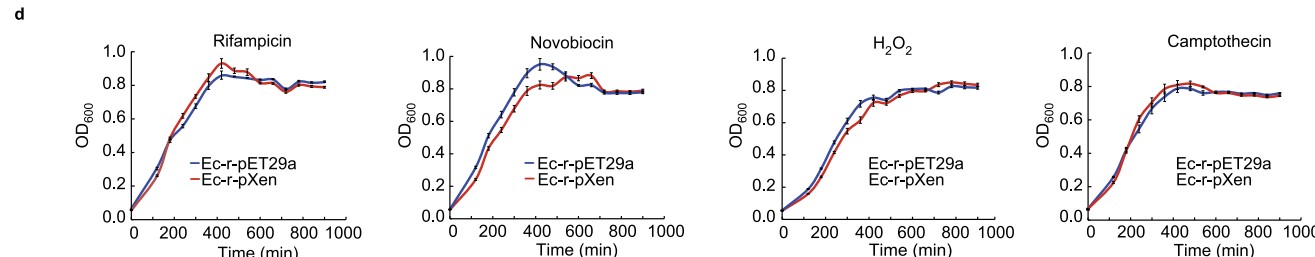

**Fig. 6 | Growth fitness of the nucleosome-forming *E. coli*, treated with 1 μM IPTG, under different nutrient sources, temperatures, pH, and chemical stress.** Comparison of colony growth for the nucleosome-forming (Ec-r-pXen) and control (Ec-r-pET29a) strains: (**a**) using solid media with different nutrients at various temperatures. M9, M9 minimal medium; M9 + Glu, M9 + Lac, and M9 + Gly represent M9 supplemented with glucose, lactose, or glycerol as carbon source, respectively; M9 + CAA, M9 supplemented with casamino acids as nitrogen source; (**b**) using LB media with different pH at various temperatures; and (**c**) using LB media with different salt concentrations at various temperatures. Data are representative of two independent experiments. Panels were seeded using serial dilutions (with 10-fold increments) of fresh bacterial culture. The figures in (**a**–**c**) are representative of two independent experiments. **d** Optical density measurement of cell growth (mean ± SEM) of the nucleosome-forming (Ec-r-pXen) and control (Ec-r-pET29a) strains at 37 °C in LB media with 1 μM IPTG, and either rifampicin (0.31 ng/mL), novobiocin (10 μg/mL), $H_2O_2$ (1.25 mM), or camptothecin (8 μg/mL). Data represent twenty-four biological replicates ($n = 24$). Source data are provided as a Source Data file.

genes, and global gene expression profiles (Fig. 4). Impressively, the transcriptome of nucleosome-forming *E. coli* remained largely unchanged throughout long-term growth. These results can be partly explained by previous studies showing the replication of nucleosome-bound DNA templates by either bacteriophage T4 DNA polymerase holoenzyme or T7 DNA polymerase and the transcription of nucleosome-bound DNA templates by either bacteriophage SP6 RNA polymerase or RNA polymerase II in vitro experiments[45–48].

Transcriptome analysis comparing the nucleosome-forming *E. coli* strain with the control *E. coli* strain identified a small number of DEGs that were mostly involved in stress response, suggesting that nucleosome formation imposes a burden on the nucleosome-forming *E. coli* strain. In preliminary experiments, we observed that elevated levels of IPTG rendered the Ec-r-pXen strain unviable. Further, the increase in nucleosome formation was not hindered by the hupA protein but caused a more severe, detrimental effect on *E. coli*. Thus, the long-term growth of nucleosome-forming *E. coli* under our experimental conditions was likely achieved through a delicate balance between cell growth and the burdens of the nucleosome. The mild phenotypic effect on nucleosome-forming *E. coli* in long-term growth is consistent with the small number of DEGs. Notably, a previous study of introducing archaeal histone-like proteins, either HMfA or HMfB, into *E. coli*, showed that the formation of archaeal histone-genome DNA complexes caused mild phenotypical changes in short-term culture (~13 h)[19]. These changes included slower growth, differences in cell length, and impaired fitness under stress in the *E. coli* strains whose HMfA or HMfB expression was induced with 15 mM rhamnose. However, the levels of archaeal histone-genomic DNA complex formation were not measured under their short-term growth conditions. Using different settings for long-term growth, we observed reduced fitness of eukaryotic nucleosome-forming *E. coli* compared to control *E. coli* in a growth competition assay and under various stress conditions, such as temperature, nutrition, pH, salt, and chemical stresses (Figs. 5, 6).

In summary, we achieved in vivo assembly of eukaryotic nucleosome core in the model bacterium *E. coli* and found that bacterial chromosomal DNA and eukaryotic histones can assemble to form nucleosome complexes that, to a certain degree, resemble those in eukaryotic cells. In addition, we have shown that nucleosome-forming bacteria can accommodate a moderate level of nucleosomes on their genome and have sustained cell division in long-term growth experiments under the condition that histone expression was induced by 1 μM IPTG treatment. The synthetic biology approach used to create the nucleosome-forming bacterium provides not only a new platform but also a unique opportunity to address the important question of how eukaryotic nucleosomes arrived during evolution. Our findings on the perceived compatibility between the eukaryotic nucleosome and the bacterial chromosome machinery have important implications for our understanding of the origin of the nucleosome and the process of eukaryogenesis.

## Methods
### Plasmid and strain construction
The histone protein-expressing plasmid pET-Xen was constructed by Gibson assembly of the codon-optimized genes encoding H2A, H2B, H3, and H4 (Supplementary Data 8) from *Xennopus laevis* synthesized by Genewiz (Suzhou, China) with the pET29a vector[49]. The H2A, H2B, H3, and H4 genes are in a polycistron with a ribosome binding sequence (RBS) placed before each gene[20]. Expression of the histone proteins is under the control of an IPTG-inducible T7 promoter (Fig. 1a and Supplementary Fig. 1a). The plasmid pET29a-amp^R was constructed by inserting the PCR fragment containing the ampicillin resistance gene and its promoter sequence, amplified from pBlue-script-SK(+) plasmid into the *Sph*I site of the pET29a plasmid by Gibson assembly. The plasmid pET28a-H2B was constructed by inserting the PCR fragment amplified from pET-Xen using primer sets 28a-H2B-F/

28a-H2B-R (Supplementary Data 9), into the *Nco*I/*Not*I site of the pET28a-TEV plasmid by Gibson assembly. The pET-Xen-H3Δ was constructed by inserting the PCR fragment, amplified from synthesized H3Δ245-310 DNA fragment by Genewiz (Suzhou, China) using primer sets H2B-seq-P5/H4-Trunc-P3 (Supplementary Data 9), into the *Sal*I/*Not*I site of the pET-Xen plasmid by T4 DNA ligation. The pET-Xen-H3del was constructed by Gibson assembly of the DNA fragment amplified from the pET-Xen using primer sets H3-K-P5/H3-K-P3 (Supplementary Data 9). The DNA fragment was first treated with *Dpn*I enzyme to remove the original pET-Xen plasmid. The plasmid pET-Xen-spGFP was constructed by Gibson assembly of fusion genes encoding H2A-GFP1-9, H2B-GFP10, H3, and H4-GFP11 (Supplementary Data 10) synthesized by Genewiz (Suzhou, China), with the pET29a vector. The pET-Xen-spGFP-H3Δ (Supplementary Data 11) plasmid was constructed by inserting the PCR fragment amplified from synthesized H3Δ245-310 DNA fragment by Genewiz (Suzhou, China), using primer sets H2B-seq-P5/H4-Trunc-P3 (Supplementary Data 9), into the *Sal*I/*Not*I site of the pET-Xen-spGFP plasmid by T4 DNA ligation.

The ancestor strains *E.coli* BL21(DE3) and *E. coli* Rosetta(DE3) used in this study were kindly provided by Dr. Yu Zhang (Shanghai, China). The strains Ec-r-pXen, Ec-r-pET29a, Ec-r-pXen-H3Δ, Ec-r-pXen-H3del and Ec-r-pET29a-amp^R were created by transforming the plasmids pET-Xen, pET29a, pET-Xen-H3Δ, pET-Xen-H3del and pET29a-amp^R into the *E. coli* Rosetta(DE3) strain respectively via the heat shock method[50]. For testing protein expression, the plasmids pET-Xen and pET29a were also transformed into the *E. coli* BL21(DE3) strain to produce strains Ec-b-pXen and Ec-b-pET29a, respectively. For testing the nucleosome assembly by visualization using split green fluorescent protein, the plasmids pET-Xen-spGFP and pET-Xen-spGFP-H3Δ were transformed into *E. coli* Rosetta(DE3) strain to produce strains Ec-r-pXen-spGFP and Ec-r-pXen-spGFP-H3Δ.

### Growth of bacteria cultures
**Bacteria culture for optimization of histone expression in *E. coli*.** To optimize the histone expression, the strains were cultured in different conditions as follows. Single colony of strains Ec-r-pXen, Ec-r-pET29a, Ec-b-pXen, or Ec-b-pET29a was individually inoculated into 3 mL LB media containing kanamycin (50 μg/mL) and chloramphenicol (25 μg/mL). The starter cultures were grown overnight at 37 °C, 220 rpm before they were amplified to 100 mL cultures of the same media and grown at 37 °C, 220 rpm, to the exponential phase ($OD_{600}$ = 0.6). Then, the cultures were added with 400 μM IPTG and continued culture at 37 °C, 220 rpm, or cooled to 18 °C before the addition of 400 μM IPTG and continued culture at 18 °C. They were harvested for protein expression assay (Fig. 1b, and Supplementary Fig. 1b, c) and ecMNase assay (see "*Methods*": "In situ micrococcal nuclease digestion assay for *E. coli* (ecMNase)") (Fig. 1c, d and Supplementary Fig. 1d) when the $OD_{600}$ reached 1.2.

**Bacteria culture and IPTG-titration experiment.** To titrate the nucleosome formation in *E. coli* with different concentrations of IPTG, single colonies of strain Ec-r-pXen and Ec-r-pET29a were picked to grow in 20 mL LB media containing kanamycin (50 μg/mL) and chloramphenicol (25 μg/mL) at 37 °C overnight. The culture was diluted to $OD_{600}$ ~ 0.1 in 500 mL of the same media and grown to an exponential phase ($OD_{600}$ ~ 0.6). They were separated into 50 mL volume cultures, and IPTG was added for a final concentration of 1, 10, 25, 50, 75, 100, 200 and 400 μM, respectively. An IPTG concentration of 0 μM was used as a control. These cells continued to be cultured with various IPTG concentrations at 37 °C for 60 min before they were harvested for protein expression assay (Fig. 3a, b), ecMNase assay with prolonged digestion, and ecMNase-seq (Fig. 3a, c).

**Long-term culture of different *E. coli* strains.** For long-term culture of the nucleosome-forming *E. coli*, Rosetta(DE3) strain freshly

transformed with pET-Xen was plated in LB plates containing kanamycin (50 μg/mL), chloramphenicol (25 μg/mL) and IPTG (1 μM), and allowed to grow at 37 °C overnight. Six colonies picked randomly from the plate were grown in LB broth supplemented with kanamycin (50 μg/mL), chloramphenicol (25 μg/mL) and IPTG (1 μM) at 37 °C on a shaker (220 rpm), before ecMNase assay was performed on them. While all six colonies were confirmed positive for nucleosome formation, four were selected for continued passages for long-term growth (Fig. 4a). In each passage, the culture was grown to plateau phase (with measured $OD_{600}$ value between 5.5 and 6.0) in about 12 h. The culture was diluted to $OD_{600}$ ~ 0.02 in 3 mL of fresh media (in a 15-mL glass tube) for the next passage and was allowed to grow under the same conditions. Such a process was repeated in passage 14. Note there were about eight rounds of cell divisions for each passage, and by estimate approximately 112 generations elapsed for passage 14. Culture samples from passages 2, 6, 10, and 14 of the four clones were collected, for which ecMNase-seq with prolonged MNase digestion, mRNA-seq, and whole genome sequencing (WGS) was performed. *E. coli* strain Rosetta(DE3) freshly transformed with pET29a (Ec-r-pET29a) was used as a control.

### SDS-PAGE analysis of histone protein expression

To detect the histone protein expression of different strains, a volume of 16.7 mL bacterial cultures ($OD_{600}$ ~ 1.2) treated with different concentrations of IPTG were harvested by centrifugation ($1844 \times g$ for 5 min at 4 °C). The cell pellets were re-suspended in 2 mL lysis buffer (50 mM Tris-HCl, pH 8.0, 500 mM NaCl, 10 mM 2-mercaptoethanol, and 5% glycerol and 0.1 mM proteinase inhibitor (PMSF)). Cells were lysed by a Scientz sonication system (Ningbo Scientz Biotechnology CO. LTD, Ningbo, China) in an ice-water bath. 100 μL of cell lysate was centrifuged at $16602 \times g$ for 15 min at 4 °C to separate soluble proteins from the cell pellet. The supernatant was treated with 20 μL 5× SDS-PAGE Protein Loading Buffer (#20315ES05, Yeasen, Shanghai). The cell pellet was suspended in 80 μL lysis buffer and 20 μL 5× SDS-PAGE Protein Loading Buffer. For total protein analysis, 80 μL of cell lysate (without centrifugation) was mixed with 20 μL 5× SDS-PAGE Protein Loading Buffer. Samples for supernatant, pellet, and total lysate (representing roughly 1.0 $OD_{600}$ bacterial culture) analysis were incubated at 95 °C for 10 min, and then centrifuged with $16602 \times g$ for 30 min at 4 °C. Samples were loaded on SDS-PAGE (15% Bis-Tris) for electrophoresis, and separated proteins were revealed by Coomassie Blue Fast Staining Solution (P0017, Beyotime, Shanghai).

### In situ micrococcal nuclease digestion assay for *E. coli* (ecMNase)

The micrococcal nuclease (MNase) digestion of genome DNA was previously used to analyze nucleosome formation in eukaryotes, which displayed a unique pattern of genomic DNA fragmentation due to nucleosome protection[22]. To make it work with the nucleosome-forming *E. coli* cells, we developed the ecMNase assay to work with *E. coli* cells by generating protoplasts from *E. coli* cells[51] applying MNase digestion to the *E. coli* protoplasts (Fig. 1c).

A volume of 16.7 mL bacterial culture ($OD_{600}$ ~ 1.2) was harvested by centrifugation ($1844 \times g$ for 5 min at 4 °C). The pelleted cells were washed with 1 mL chilled buffer B (25 mM Tris-HCl, pH 8.0, 456 mM sucrose). Following centrifugation at $1844 \times g$ for 5 min at 4 °C, bacterial cells were incubated in 2 mL buffer A (25 mM Tris-HCl, pH 8.0, 456 mM sucrose, 0.042% EDTA pH 7.0, and 0.21 mg/mL lysozyme) for 17 min at 25 °C to generate protoplasts. The protoplasts were pelleted ($1844 \times g$ for 5 min at 4 °C) and washed with buffer B three times. Then the protoplasts were re-suspended in 2.0 mL MNase buffer (1.0 M sorbitol, 50 mM NaCl, 1 mM $CaCl_2$, 10 mM Tris-HCl pH 7.4, 5.0 mM $MgCl_2$, 1.0 mM beta-mercaptoethanol, 0.5 mM spermidine and 0.1% Nonidet NP-40). Aliquots of 250 μL protoplast suspension (representing 2.5 OD of bacterial culture) were treated with 200 unit/mL MNase

(N863776-10KU, Maklin, Shanghai) at 37 °C for different lengths of time, i.e., 0.5, 1, 2, 3, 4, 6 and 8 min, respectively, before reactions were stopped by addition of 12.5 μL 0.5 M EDTA to each reaction. For prolonged MNase digestion, 200 units/mL MNase and 12 min digestion time were used. Each aliquot was further treated with 0.5% SDS, 100 μg/mL ribonuclease A, and 200 μg/mL proteinase K at 37 °C for 1 h. DNA fragments were precipitated with ethanol and re-suspended in 20 μL distilled water before they were assessed by agarose DNA electrophoresis (2% agarose gel in 1 × TAE).

### Core histone protein expression in *E. coli* and nucleosome purification

The core histone proteins were expressed, and the nucleosome complexes were purified based on the previously developed method described[20]. The Ec-r-pXen cells were grown at 37 °C to $OD_{600}$ ~ 0.6, and induced with 400 μM IPTG for 16 h. Cells were harvested by centrifugation at $4722 \times g$ for 5 min from a 6-liter LB culture and re-suspended in 100 mL of lysis buffer containing 50 mM Tris-HCl (pH 8.0), 500 mM NaCl, 5% glycerol, and 1 mM DTT. The cells were then disrupted by six rounds of sonication at 4 °C using an Avestin Emulsiflex-C5 cell disrupter (Avestin, Inc.). The supernatant was collected by centrifugation at $18882 \times g$ for 1 hr at 4 °C, mixed with 2 mL of HisSep Ni-NTA Agarose Resin (Yeasen, Shanghai), and rotated for 1 hr at 4 °C. The mixture was then loaded on a 60 mL column, and the resin was washed with 20 mL of lysis buffer containing 10 mM imidazole before being eluted with 70 mL of lysis buffer containing 40 or 300 mM imidazole, respectively. The fraction that eluted with buffer containing 300 mM imidazole was concentrated up to 1 mL by ultra-filtration using an ultrafiltration centrifugal tube (15 mL/3 KD, Millipore, Inc.) at 4 °C and then injected into a Superdex 200 increase 10/300 GL column (GE Healthcare, Inc.) for further purification. The histone octamer peak was eluted at an elution volume of 10 mL, and the peak fractions were analyzed using 15% SDS-PAGE.

### Atomic force microscopy (AFM) imaging

*E. coli* cells were grown at 37 °C to $OD_{600}$ ~ 0.6, and then induced with 75 μM IPTG for 1 h until reaching OD600 ~ 1.2. Subsequently, 16.7 mL bacterial culture was harvested by centrifugation at $1844 \times g$ for 5 min at 4 °C. Protoplasts were generated as described in *Methods*: "In situ microccocal nuclease digestion assay for *E.coli* (ecMNase)". After that, the protoplasts were re-suspended in 2.0 mL MNase buffer to release the DNA-nucleosome complexes, and centrifuged at $16602 \times g$ for 30 min at 4 °C. Next, the supernatant was diluted 1:100 and deposited onto a freshly cleaved mica substrate treated with 3-aminopropyltriethoxysilane (APTES), before it was subjected to AFM under ambient air conditions using Bruker Dimension IconTM (Bruker). AFM was operated in the tapping modes using the silicon cantilever, SNL-10 (Bruker, US), with a typical resonance frequency of 50–80 kHz and a spring constant of 0.175 N/m. The normal scanning rate was 1.76 Hz. The images were captured in a 996.1 × 996.1 nm format and flattened and plane-fitted before analysis.

The nuclei of *Xenopus* liver cells were isolated following the method as previously described[52]. Briefly, 1 g of *Xenopus* liver was suspended in 10 mL of buffer containing 2.0 mM $MgCl_2$, 2.0 mM beta-mercaptoethanol, 0.5 mM spermine, and 10 mM HEPES pH7.5. The liver suspension was homogenized using a glass homogenizer, and the resulting homogenate was centrifuged at $1844 \times g$ for 10 min to pellet the nuclei. The isolated nuclei were then lysed to liberate nucleosomes using the same procedure as that for the protoplasts of *E. coli* cells before AFM imaging analysis.

### ecMNase sequencing (ecMNase-seq) of the nucleosome-forming *E. coli*

A volume of 16.7 mL Ec-r-pXen culture ($OD_{600}$ ~ 1.2) was treated with ecMNase assay using prolonged digestion conditions. Gel slices

containing genomic fragments corresponding to the mono-nucleosome range (approximately 100 to 180 bp) were cut, and genomic fragments were extracted for Illumina sequencing. Extracted DNA was quantified and sequenced using the Illumina Hiseq X Ten instrument according to the manufacturer's instructions (Illumina, San Diego, CA). Note a reduced number of PCR cycles (~4 cycles) were used to ensure the minimum redundancy of ecMNase-seq data. Illumina sequencing was performed by Genewiz (Suzhou, China). Sequencing was carried out using a 2 × 150 paired-end (PE) configuration.

For titration of nucleosome assembly in Ec-r-pXen with varying IPTG concentrations, ecMNase-seq was performed with two biological replicates at each IPTG concentration. For the long-term growth experiment of the nucleosome-forming *E. coli*, ecMNase-seq was performed at each passage (2, 6, 10, and 14) for four different clones. Sample data information is included in Supplementary Data 12.

### Transcriptome sequencing (mRNA-seq) and whole genome sequencing (WGS) analysis of *E. coli* in long-term culture

A volume of 15 mL Ec-r-pXen culture ($OD_{600}$~1.2) was harvested by centrifugation (1844 × *g* at 4 °C for 15 min). Total RNA was extracted from each sample using TRIzol Reagent (Thermo Fisher Scientific, 15596026). The total RNA purity and yield were checked using Agilent 2100 Bioanalyzer (Agilent Technologies, Palo Alto, CA) and NanoDrop 2000 (Thermo Fisher Scientific, USA). For each sample, 1 μg total RNA with a RIN value above 6.5 was used for the preparation of the Illumina library. rRNA was first depleted from total RNA using an rRNA removal Kit (D5.VAHTS Total RNA-seq (Bacteria) Library Prep Kit for Illumina, Vazyme). Illumina libraries were made using the rRNA-depleted RNA samples by Genewiz (Suzhou, China) using the manufacturer's protocol (Illumina, San Diego, CA). Note a reduced number of PCR cycles (~13 cycles) was used to ensure the minimum redundancy of RNA-seq data. Illumina sequencing was performed by Genewiz (Suzhou, China) using the Illumina Hiseq X Ten instrument according to the manufacturer's instructions (Illumina, San Diego, CA). Sequencing was carried out using a 2 × 150 paired-end (PE) configuration. mRNA-seq was performed at each passage (2, 6, 10, and 14) for four different clones of Ec-r-pXen, and for two different clones of Ec-r-pET29a. mRNA-seq was performed at each passage (2, 6, 10, and 14) for four different clones of Ec-r-pXen, and for two different clones of Ec-r-pET29a. The mRNA-seq data information is included in Supplementary Data 13.

For WGS, genomic DNA was isolated from *E. coli* culture (OD ~1.2) for long-term culture experiments using a DNA isolation kit (Bacteria DNA Kit, Omega Inc). Genomic DNA was used for Illumina library preparation using the VAHTS Universal DNA Library Prep Kit (7E481D0, Vazyme, Nanjing) following the manufacturer's protocol. Illumina sequencing was performed by Genewiz (Suzhou, China) using the Illumina Hiseq X Ten instrument according to the manufacturer's instructions (Illumina, San Diego, CA). Sequencing was carried out using a 2 × 150 paired-end (PE) configuration. The WGS data information is included in Supplementary Data 14.

### LC-MS/MS analysis for proteome from strains Ec-r-pXen and Ec-r-pET29a

A volume of 1.53 mL Ec-r-pXen ($OD_{600}$~1.3) culture and 1.67 mL Ec-r-pET29a ($OD_{600}$~1.2) culture grown in LB medium containing 1 μM IPTG, was harvested by centrifugation at 1844 g for 5 min and re-suspended in 80 μL of lysis buffer. Then, 20 μL of 5 × SDS-PAGE Protein Loading Buffer was added. The mixtures were incubated at 95 °C for 10 min, vortexed vigorously for 3 × 30 s, and centrifuged at 16602 × *g* for 30 min at 4 °C. Next, 10 μL of the supernatant were loaded onto a 15% Bis-Tris SDS-PAGE for electrophoresis. When the protein samples reach the interface between the stacking gel and the separating gel, stop the electrophoresis, and excise the protein bands from the gel.

The gel slides of Ec-r-pXen and Ec-r-pET29a were cut into 1 mm particles and treated with 50% acetonitrile for decolorization, followed

by dehydration with 100% acetonitrile. Proteins in the particles were reduced with 10 mM TCEP and 50 mM CAA at 37 °C for 30 min. Next, all samples were digested with 400 ng of trypsin (Promega, Madison, Wisconsin) at 37 °C overnight. The resulting digested samples were then acidified with 2 M HCl, cleaned up using C18, and dried using a vacuum centrifuge. The peptides were then suspended in 50 μL solution containing 70% ACN, 0.1% FA, and 98% $H_2O$.

Samples were analyzed using the Thermo Orbitrap Fusion Lumos coupled to the EASY-NIC 1200 (Thermo Fisher Scientific). The peptides were injected (1 μL) onto a C18 trap column (ID × OD × L, 20 μm × 360 μm × 50 mm) (P/N 6041.5290; OMIC SOLUTION). Mobile phase A was 0.1% formic acid in water, and mobile phase B was 80% ACN/0.1% formic acid. Separations were performed at 600 nL/min across a 50 min linear gradient from 3 – 100% B. Each survey scan was acquired in the Orbitrap at 60,000 FWHM with the scan range from 375–1500 m/z.

### Fluorescence imaging for morphological analysis

Bacteria cultures were grown to $OD_{600}$~0.7 in LB broths. 1.43 mL of culture was harvested by centrifugation at 1844 × *g* for 3 min. Pellets were washed with PBS buffer (0.137 M Sodium chloride, 0.0027 M Potassium Chloride, 0.01 M Sodium Phosphate Dibasic and 0.0018 M Potassium Phosphate Monobasic, pH 7.4), and then stained with 50 μg/mL DAPI (4′,6-diamidino-2-phenylindole, Sigma, German) for 10 min on ice. 1 μL of stained cellular suspension was spread onto a prepared 1% agarose pad on slides with excess liquid removed. Slides were visualized, and photos were taken using a Leica TCS SP8 STED 3 × confocal laser scanning microscope with an HC PL APO CS2 40 × /1.30 oil objective and a HyD1 detector. The excitation and emission wavelengths for DAPI were 405 nm and 405–480 nm, respectively. The bright field was detected with the PMT detector. Images were analyzed using the Image-Pro Plus 6.0 software (Media Cybernetics, Silver Spring, MD).

### Fluorescence imaging for visualization of nucleosome assembly using split green fluorescent protein

Cells were grown to OD600 ~0.6 at 37 °C and induced with 400 μM IPTG for 1 hr till $OD_{600}$~1.2. Then, 0.83 mL of culture was harvested by centrifugation at 1844 × *g* for 3 min. Cell pellets were washed with PBS buffer and then stained with DAPI (50 μg/mL, Sigma, German) for 10 min on ice. 1 μL of stained cellular suspension was spread onto a prepared 1% agarose pad on slides with excess liquid removed. Slides were visualized, and photos were taken using the Nikon AX R confocal with a 60 × oil-immersion objective of 1.42 NA. The excitation and emission wavelengths for GFP were 488 nm and 527–551 nm, respectively. The excitation and emission wavelengths for DAPI were 405 nm and 429–474 nm, respectively. Images were analyzed using the Fiji.

### Growth competition assay for mixed *E. coli* strains

Two strains, Ec-r-pXen and Ec-pET29a-amp$^R$ were used for growth competition assay. An equal number (1 × 10$^5$) of Ec-r-pXen and Ec-pET29a-amp$^R$ cells from exponentially growth stage ($OD_{600}$~0.4) were inoculated into the same LB medium (20 mL) containing IPTG (1 μM) and grew in a 100-mL flask with shaking (220 rpm) at 37 °C. The mix-culture was diluted to $OD_{600}$ = 0.02 every 12 h with fresh media containing IPTG. The mix-cultures at time points of 0, 12, 24, 36, 48, 60, and 72 h, were plated on solid LB plates containing kanamycin (50 μg/mL). 100 colonies (kanamycin positive) grown on the kanamycin plates were then transferred to LB plates containing both kanamycin (50 μg/mL) and ampicillin (50 μg/mL). To measure their growth fitness, the number of double positive clones (kanamycin and ampicillin positive; representing Ec-pET29a-amp$^R$) were compared to the number of single positive clones (kanamycin; representing Ec-r-pXen).

### Stress test under varying nutrients, pH, and salt concentrations

Solid media on plates used in this assay include Luria-Bertani (LB) (Tryptone 1%, Yeast extract 0.5%, NaCl 1%, pH 7.0), LB with NaCl at 0.5%, 1%, 3% and 5%, LB with pH 5.0, pH 7.0 and pH 9.0, R2A medium (Yeast extract 0.5 g/L, Difco Proteose Peptone no. 3 0.5 g/L, Casamino Acids 0.5 g/L, Glucose 0.5 g/L, Soluble starch 0.5 g/L, Sodium pyruvate 0.3 g/L, $K_2HPO_4$ 0.3 g/L, $MgSO_4 \cdot 7H_2O$ 0.05 g/L), M9 minimal medium (9 mM NaCl, 22 mM $KH_2PO_4$, 47.8 mM $Na_2HPO4$, 19 mM $NH_4Cl$, 2 mM $MgSO_4$, 0.1 mM $CaCl_2$, 0.4% glucose), M9 + CAA medium (19 mM $NH_4Cl$ substituted with 4% casein amino acid), M9 + Gly medium (0.4% glucose substituted with 0.4% glycerol), M9 + Lac medium (0.4% glucose substituted with 55.5 mM lactose) and M9-Glu (0.4% glucose substituted with 2% glucose). The addition of 0.4% glucose in M9 did not inhibit IPTG induction of protein expression in *E. coli* lambda DE3/ pET system[53].

To start the assay, fresh bacterial cultures in log-phase were diluted to $OD_{600} = 0.5$, and serial dilutions were made in 10-fold increments before plated onto the different types of solid media plates. Culture plates were incubated at appropriate temperatures and monitored for growth. The formation of colonies was recorded by taking photos using a NIKON D810 camera at different time points (Supplementary Data 15) for Fig. 6a, b.

### Stress test using various chemical agents

For the nucleosome-forming *E. coli* and control strain containing empty plasmid, their seed cultures of 3-mL LB media supplemented with antibiotics kanamycin (50 μg/mL), chloramphenicol (25 μg/ mL) and IPTG (1 μM) were grown overnight at 37 °C. To measure the growth rate of different *E. coli* strains, they were diluted to $OD_{600}$ 0.02 in 20 mL fresh media. 200 μL of fresh dilute were plated in replicates into a flat bottom Nunc 96-well plate (ThermoFisher Scientific, UK) and incubated in a microplate shaker (800 rpm) at 37 °C. Optical density was measured at 600 nm with a photometer Multiscan Go (Thermo Fisher Scientific, USA) every 60 min for 15 h.

For assaying growth under chemical stress, the chemicals, i.e., novobiocin (final concentration 10 μg/mL, TargetMol, USA), rifampicin (0.31 ng/mL, LEAGEME, China), camptothecin (8 μg/mL, MedChemExpress, USA), and hydrogen peroxide ($H_2O_2$ 1.25 mM, Mkbio, China), were each added to the fresh culture media to the final concentration as indicated. They were grown using the same protocol for culture in 96-well plates using a microplate shaker, and their optical density was measured at 600 nm with a photometer Multiscan Go every 60 min for 15 h.

### ecMNase-seq data processing and peak calling analysis

Paired-end ecMNase-seq reads were first trimmed for adapter sequence using Cutadapt (version 2.7)[32] and then merged using BBmerge (version 38.26)[36] with default parameters. The merged reads between 100 and 180 bp in length were considered to be mono-nucleosomes. For nucleosome peaks called at various IPTG concentrations for Fig. 3e–g, reads for each IPTG concentration samples of two biological replicates were pooled, and down-sampled with random drawing using the quantities in proportion to those of the titration results from Fig. 3d. For nucleosome peaks called in long-term culture, reads for each sample were down sampled to the same number ($n = 3000000$). The sampled reads were then mapped onto the *E. coli* BL21(DE3) reference genome (CP001509.3) using Bowtie2 (version 2.2.6) (parameters: --local -X 2000). The resulting alignments, recorded in the SAM file, were converted to a sorted bam file and filtered for reads with mapping quality ≥ 20 using Samtools (version 1.4). The processed mapping files were then inputted to Danpos2 (version 2.2.2)[36] to do nucleosome peak calling (parameters: -jd 120 -n N -u 1e-10) and nucleosome profiling analysis.

### mRNA-seq data processing and differential expression analysis

mRNA-seq reads were first trimmed using Cutadapt (version 2.7)[32] to remove adapter sequences. Low-quality bases were trimmed using Trimgalore (version 0.6.4) (https://www.bioinformatics.babraham.ac. uk/projects/trim_galore) (parameter: -q 30). The cleaned reads were then mapped onto the *Escherichia coli* BL21(DE3) reference genome (CP001509.3) using Bowtie2 (version 2.2.6) (parameters: --local −3 5 -X 2000). Samtools (version 1.4) was used to convert the mapping sam file to a sorted bam file with the "sort" function. Read counts for genes were computed from read alignments using the summarizeOverlaps function (parameters: singleEnd = FALSE, ignore.strand = TRUE, fragments = TRUE) of the R package GenomicAlignments (version 1.22.0)[54]. Normalization of counts (in transcripts per million (TPM) is indicated in each figure or figure legend. Differential gene expression (DEGs) analysis was carried out using the R package DESeq2 (version 1.28.0)[55] with the default Wald test and the Benjamini-Hochberg multiple testing correction. The DEGs were identified by DEseq2 with a screening threshold of fold change ≥ 2 and *q*-value < 0.001.

### Whole genome sequencing data processing and mutation analysis

Paired-end FASTQ files were processed with the following pipeline. First, Trimmomatic[56] (version 0.38) was used to remove adapters, reads shorter than 50 bp, and poor quality reads (parameters: LEADING:3 TRAILING:3 SLIDINGWINDOW:10:30 HEADCROP:5 CROP:140 MINLEN:50). Data quality was assessed using FastQC (version 0.11.8). Processed reads were aligned to the *E. coli* BL21(DE3) reference genome (CP001509.3) using BWA (version 0.7.17) mem algorithm, and Sam files were converted to sorted and duplication-removed Bam files using Samtools (version 1.4) using "sort" and "rmdup". Variants were called by bcftools (version 1.9) using "mpileup" (parameters: -q 20 -Q 20), "call" (parameters: --ploidy 1 -m -Ov), and "filter" (parameters: -s FILTER -g 10 -G 10 -i "%QUAL > 20 && DP > 10 && MQ > 40 && (DP4[2] + DP4[3]) > 4 && (DP4[2] + DP4[3])/DP > 0.1"). Samtools (version 1.4) "depth" was used to determine the read coverage for the genome.

### Nucleosome profiling and correlation analysis

For the nucleosome peak density profiling cross the *E. coli* genome in IPTG-titration experiments (Fig. 3f), the nucleosome peak density was computed using a step size of 5 kb and a span size of 25 kb. The violin plots presented in Fig. 3f were displayed with a y-axis limit set from 80 to 160. For the dyad nucleosome occupancy profiling of the nucleosome-forming *E. coli* from passages 2, 6, 10, and 14 (Fig. 4c), the nucleosome occupancy was analyzed within a window spanning 200 bp on the upstream and downstream side of the peak centers, and the dyads were plotted in the same order based on their occupancy ranking in passage 14. The occupancy heatmap and averaged profile plots were performed with deeptools (version 3.4.3)[57] using the "plotHeatmap" function. For the nucleosome profiling around the transcription start sites (TSS), the averaged dyad frequency (Fig. 4e) was analyzed within a sliding window (size = 50 bp) spanning − 500 to + 1000 bp flanking TSS, and was plotted with the "geom xspline" function (parameters: spline shape = 0.5) in R package ggalt (version 0.4.0). For the nucleosome profiling around the TSS of an individual gene in Fig. 4f, nucleosome occupancy was analyzed within a sliding window (size = 50 bp) spanning from − 500 bp to + 1000 bp around the TSS, and the profiling line was plotted with the "spline connected" function in Origin software (version 8.0951). We defined − 1 and + 1 nucleosome positions as relative to the upstream and downstream of TSS, respectively.

Correlation analysis was measured by Pearson's correlation coefficient with "cor" from R (version 3.6.0). For Fig. 3f, the peak density distribution was pair-analyzed for each IPTG treatment. For Fig. 4d, the nucleosome occupancy between corresponding dyads was analyzed for each pair of passages.

## Statistical and hierarchical clustering analysis

Statistical analyses were conducted using SigmaPlot (version 14.0) and were specifically indicated in figure legends. For data distribution conferring to non-normality, a statistical comparison was performed with the non-parametric Mann-Whitney Rank Sum Test ($P < 0.05$ was considered statistically significant).

For Fig. 4g, the heatmap was plotted with the R package pheatmap (version 1.0.12), and hierarchical cluster analysis was performed with the 'cluster_cols' function in pheatmap based on Euclidean distance using the 'complete' method.

## Reporting summary

Further information on research design is available in the Nature Portfolio Reporting Summary linked to this article.

## Data availability

All sequence data generated in this study have been deposited in the NCBI database under accession code BioProject ID PRJNA854336, and listed in Supplementary Data 12–14. The LC-MS/MS raw data for Ec-r-pXen and Ec-r-pET29a strains generated in this study have been deposited in iProx under accession code IPX0009407000. Source data are provided in this paper.

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

## Acknowledgements

This work was supported in part by the National Natural Science Foundation of China (32393971 awarded to X.J., 92451303 and 32270719 awarded to X.L., 32200093 awarded to P.C.), the National Key R&D Program of China (2023ZD04073 awarded to X.L.), the National Science and Technology Major Projects (2018YFA0903700 awarded to X.J., 2019YFA0904600 awarded to Yan Zhu), and the Strategic Projects of the Chinese Academy of Sciences (XDA24010403 awarded to X.L.). We thank Fan Gong at the National Facility for Protein Science in Shanghai (NFPS), Shanghai Advanced Research Institute, CAS, for technical support with AFM experiments, and Yuan Yuan Gao, Shanshan Wang, Lianyan Jing, and Xiaoyan Xu at the core facility of the Center for Excellence in Molecular Plant Sciences (CEMPS) for assistance with LC-MS/MS experiments.

## Author contributions

X.L. conceived the project and wrote the manuscript. X.J., X.Z., N.Z., P.C., J.G., K.Z., X.W., and W.C. performed experiments, analyzed data, and/or prepared the manuscript. B.C.Y, P.H., G.Z., and S.Y. advised on the study, and/or helped with data analysis.

## Competing interests

The authors declare no competing interests.
