## [Peer Review File · Nature Communications]

Reviewers' Comments:

Reviewer #1:

Remarks to the Author:

We respect the author's persistence in getting this published. At the heart of it, this manuscript shows that nucleosome-like structures (consisting of the four histones) can be formed in *E. coli*, and that *E. coli* is viable when these histones are expressed at a very low level. This per se isn't new, as people have expressed all four histones in a polycistronic messenger for isolation of histone octamers before, and as correctly pointed out by the authors. (Shim et al, 2012). However, the IPTG concentration for that paper was only marginally lower than here. I don't see how the authors can say 'we successfully engineered the in vivo assembly of the nucleosome core in the model bacterium *E. coli* and showed that the host chromosomal DNA can be folded to form nucleosome complexes with many features resembling those found in eukaryotic cells'. This is an overstatement and has to be toned down. What did they engineer that wasn't already in Shim et al? Also, there is no evidence that the host chromosome is 'folded', as in forming higher order structure. All they can say is that some nucleosome-like structures are assembled when histones are expressed.

As such, the impact of the manuscript is rather moderate, however I believe that there might be some interest in this work. However, we ask the authors to further tone down their claims of 'chromosome folding', to avoid misleading the reader, and also should remove other misleading statements as indicated.

Abstract:

- The chromosome is not folded (no evidence).
- What is a 'stabilized nucleosome?' stabilized compared to what? A stable nucleosome?
- Calling low levels of histone expression 'moderate' doesn't change things. The histone levels that allow viability are very low.

Main text

- The AFM is not convincing. The control (*xenopus* liver) also looks remarkably depleted in nucleosome, and it is not clear what the Ecr-r-pXEN panel shows. The figure legend for i) could be more clear (i.e. explain what 'Xenopus' is (e.g. chromatin isolated from *xenopus laevis* liver). Please also do a height analysis of the control (middle panel).
- For context, state clearly at which IPTG concentrations the first experiments were done.
- All growth experiments were done at 1 uM, where there are no growth defects (and the

competition experiment is nice). I like the IPTG titration, but not enough detail is given. However, it appears (but one really has to look for this information) that the majority of experiments were done at this low concentration, apart from the IPTG titration experiment. Fig. 3A should include time after IPTG induction.

- It would be interesting to describe what the effect on growth is of higher IPTG concentrations. Do they die eventually? grow slower?
- Entire gels for MNase gels should be shown (at least in sup figs). The author's argumentation that because it is the 'same' it can be cut off is not valid, especially when all the action is very close to where the gel is cut off (e.g. 3c).
- Fix grammar in line 130 (as efficient)
- Discussion: 'low' was changed to 'moderate' but levels are the same.
- Discussion line 397 – overstated
- Line 414 – overstated ('low' nucleosome).

Point-by-point Responses to Reviewers' Comments

Manuscript ID: NCOMMS-24-32792A

Creating a eukaryotic nucleosome-forming bacterium

Please note responses to reviewer' comments are in blue text; the cited references are listed at the end of the point-by-point response.

REVIEWER COMMENTS

Reviewer #1 (Remarks to the Author):

We respect the author's persistence in getting this published. At the heart of it, this manuscript shows that nucleosome-like structures (consisting of the four histones) can be formed in *E. coli*, and that *E. coli* is viable when these histones are expressed at a very low level. This per se isn't new, as people have expressed all four histones in a polycistronic messenger for isolation of histone octamers before, and as correctly pointed out by the authors. (Shim et al, 2012). However, the IPTG concentration for that paper was only marginally lower than here. I don't see how the authors can say 'we successfully engineered the in vivo assembly of the nucleosome core in the model bacterium *E. coli* and showed that the host chromosomal DNA can be folded to form nucleosome complexes with many features resembling those found in eukaryotic cells'. This is an overstatement and has to be toned down. What did they engineer that wasn't already in Shim et al? Also, there is no evidence that the host chromosome is 'folded', as in forming higher order structure. All they can say is that some nucleosome-like structures are assembled when histones are expressed.

As such, the impact of the manuscript is rather moderate, however I believe that there might be some interest in this work. However, we ask the authors to further tone down their claims of 'chromosome folding', to avoid misleading the reader, and also should remove other misleading statements as indicated.

We are grateful to your great effort in reviewing and helping improve our manuscript, thank you very much!

We hope to stress that we have fully credited Shim *et al's* (2012) work on co-expression of the *Xenopus* core histones in *E. coli*, although *in vivo* assembly in *E. coli* was not attempted in their work -- it was not known before our current study whether the assembly of the higher-order structure of the eukaryotic nucleosome, i.e., the octasome-DNA complex, can be achieved *in situ* on bacterial genome DNA.

We used multiple-lines of experiments (data in Figs. 1 and 2) to collectively demonstrate the successful *in situ* assembly of eukaryotic nucleosomes on the bacterial genome. After titration experiments with varying IPTG concentrations (1~400 μ M; Fig. 3), we used a condition (1 μ M IPTG) to induce a certain level (described as 'moderate' - please see data discussion in the following questions below) of nucleosome-formation in *E. coli*, which saw a sustained growth of the nucleosome-forming strains for at least 110 divisions in longer-term growth experiments. The long-term viability of nucleosome-forming bacterium may reflect a delicate balance between cell growth and the burden of the nucleosomes.

These points are further addressed in details in the following related questions, with many changes made to address them. Thank you!

Abstract:

- The chromosome is not folded (no evidence).

Response:

In the original texts “We show that bacterial chromosomes can be folded *in vivo* to form nucleosome complexes with many features resembling those found in eukaryotes”, the word “folded” was used to describe “bacterial DNA wraps around nucleosomes” in *E. coli* cells. (The formation of nucleosome complexes on bacterial chromosome DNA was supported by data of Figs. 1 and 2).

To avoid confusion by the word “fold”, we have revised the sentence to:

“We show that bacterial chromosome DNA and eukaryotic histones can assemble *in vivo* to form nucleosome complexes with many features resembling those found in eukaryotes.” (page 2, lines 31-32, blue text).

Some other occurrences are also updated (page 5, lines 98-99; page 17, lines 407-408; page 19, lines 464-465; blue text). Thank you very much for carefully examining our wording!

- What is a ‘stabilized nucleosome?’ stabilized compared to what? A stable nucleosome?

Response:

We have replaced ‘stabilized’ with “stable” in the abstract and in a number of other places (page 2, line 37; page 13, line 304; page 18, line 431; page 45, line 1130; blue text).

In the original texts “It exhibits stabilized nucleosome formation, a consistent transcriptome across passages, and reduced growth fitness under stress conditions”, the word “stabilized” was intended to describe “a stable nucleosome presence in *E. coli* between the passages in the long-term growth experiments. The use of ‘stable’ would be more appropriate. Thanks!

- Calling low levels of histone expression ‘moderate’ doesn’t change things. The histone levels that allow viability are very low.

Response:

We are thankful for the comments, which led us to rethink and reflect on our data and results.

We also appreciate your patience in going through the re-analyzing process with us! Please allow us to explain our endeavor/struggle in reaching our conclusion --

between using ‘low’ or ‘moderate’, we took a computational approach and re-analyzed the data from different data levels (please see details below).

We concluded that ‘moderate’ is appropriate, basing on estimation using the transcriptomics and the proteomics data, as well as using ecMNase data from the IPTG titration experiments.

- 1) At the transcriptomics level, the abundance of histones’ poly-cistron transcript in the nucleosome-forming *E. coli* (with 1 μ M IPTG treatment) was ranked on average 53rd out of 4160 gene transcripts (top 12.74 percentile) in the long-term growth experiment.

In comparison, several *E. coli* proteins that are involved in transcription and DNA replication processes (i.e., RNA polymerase alpha subunit, DNA polymerase subunits: alpha, beta, and epsilon, and DNA-binding protein HU subunits: alpha and beta), were ranked 15th, 1024th, 374th, 1333rd, 1378th and 3106th, respectively (Supplementary Data 14) (page 12, lines 285-291, blue text).

- 2) At the proteomics level, the abundance of histones’ proteins was determined by LC-MS/MS analysis on the proteome of the nucleosome-forming strain that underwent long-term growth (with 1 μ M IPTG treatment). The protein abundance of histone H2A, H2B, H3 and H4 were estimated using emPAI (Exponentially Modified Protein Abundance Index), which were ranked 542nd (top 36.30 percentile), 578th (top 38.71 percentile), 202nd (top 13.53 percentile), and 144th (top 9.65 percentile) among 1493 identified proteins, respectively.

In comparison, the *E. coli* proteins involved in transcription and DNA replication processes (i.e., RNA polymerase alpha subunit, DNA polymerase subunits: alpha, beta, and epsilon, and DNA-binding protein HU subunits: alpha and beta), had emPAI estimates that were ranked 40th, 1242nd, 419th, 530th, 143rd and 457th among 1493 identified proteins (the same set of identified proteins), respectively (Supplementary Data 16) (page 15, lines 353-360, blue text).

- 3) The relative levels of nucleosome-formation in *E. coli* in the titration experiments, were estimated using the ecMNase assay by comparing the amount of mono-nucleosome protected DNA under various IPTG concentrations. Basing on the Fig. 3 data, we found that nucleosome formation at 1 μ M IPTG was about 13.2% of that at 50 μ M IPTG, and 7.4% of that at 400 μ M IPTG (Notably at 400 μ M IPTG level, high over-expression of foreign proteins is often achieved using the pET system in BL21 cells^{1,2}). These data indicated significant nucleosome formation in *E. coli* with 1 μ M IPTG treatment (page 18, lines 424-427, blue text).

Taken together, the levels of histones expression are in the top 12.74 percentile in *E. coli* transcriptome, and in the top 9.65 to 38.71 percentile range in *E. coli* proteome. Nucleosome formation at 1 μ M IPTG is about 13.2% of that at 50 μ M IPTG, and 7.4% of that at 400 μ M IPTG. So, we conclude that ‘moderate’ is appropriate basing on the estimation from above discussion -- We thus stay with using “moderate” to describe the nucleosome-formation in the longer-term growth experiments throughout the manuscript.

(We hope to clarify that this condition (1 μ M IPTG) for longer-term growth of nucleosome-forming *E. coli* was carefully chosen after preliminary tests -- it reflects a delicate balance between cell growth and the burden of the nucleosomes. Please see further discussion in other related questions below).

And again, thank you very much for helping us rethink and reflect on our study!

Main text

- The AFM is not convincing. The control (xenopus liver) also looks remarkably depleted in nucleosome, and it is not clear what the Ec-r-pXEN panel shows. The figure legend for i) could be more clear (i.e. explain what 'Xenopus' is (e.g. chromatin isolated from xenopus laevis liver)). Please also do a height analysis of the control (middle panel).

Response:

We are thankful for the comments that help us improve on our AFM data! We have updated figure legends to explain with more details on the AFM experiments, and provided new height data to support our results. The AFM experiments were performed to provide one important piece of evidence, among many, to support the assembly of nucleosome core on *E. coli* genome DNA - the AFM results corroborate with other data to support the formation of nucleosome core in *E. coli*.

We have updated Fig. 1i (with light blue arrows) and its legends (blue text), results (page 7, lines 162-164, blue text), and methods (page 27, lines 651-652, blue text) to explain with more details on the AFM data:

- 1) For the negative control (Ec-r-pET29a, middle panel) and the nucleosome-forming *E. coli* (Ec-r-pXen, right panel), their protoplasts were first prepared from *E. coli* cell as described (*Methods*), and then lysed with MNase buffer (containing sorbitol, beta-mercaptoethanol, spermidine and Nonidet NP-40, etc.) to release chromosome DNA or DNA-nucleosome complexes as described in *Methods*. Chromosome DNA (for Ec-r-pET29a cells) or DNA-nucleosome complexes (for Ec-r-pXen cells) was further diluted, and deposited onto mica substrate and subjected to AFM analysis.
- 2) For the positive control (left panel), the nuclei were isolated from *Xenopus* liver cells as described previously³. Nucleosomes were liberated by being subjected to the same procedure as that used for *E. coli* protoplasts. DNA-nucleosome complexes from lysed nuclei were deposited onto mica substrate and subjected to AFM analysis.

Obtaining good AFM images from native (*Xenopus* and *E. coli*) samples is extremely challenging and time-consuming. We tried and made a series of dilutions (1:100 to 1:10000) of the DNA-nucleosome suspension in order to obtain dispersed single DNA fiber. The positive control (Fig. 1i, left panel) prepared from *Xenopus* liver samples, showed beads-on-a-string like structures along a DNA strand. The nucleosomes are unevenly distributed along DNA, and some regions are less populated. We found that

similar profiles were shown in others' preparation of *Xenopus* or Human's nucleosomes for AFM analysis^{4,5} (please see below: Rsp Fig. 1 and Rsp Fig. 2).

Rsp Fig. 1. AFM image showing *Xenopus* nucleosomes along DNA strand⁴.

Rsp Fig. 2. AFM image showing human nucleosomes along DNA strand⁵.

Height analysis was performed on the chromosome DNA of the negative control (middle panel), which is shown below (Rsp Fig. 3). It has been added to Supplementary Figure 1f (see 'Supplementary Information', page 2). Notably, the height values for negative control match those for chromosomal DNA^{6,7}, which are significantly smaller than those of the nucleosome particles from *Xenopus* liver cells and from Ec-r-pXen cells (Rsp Fig. 4).

Rsp Fig. 3. Histogram diagram of DNA strand height for Ec-r-pET29a (n=31). It has been added to Supplementary Fig. 1f ('Supplementary Information', page 2).

Rsp Fig. 4. Histogram diagram of the bead's height in the beads-on-a-string structure for Ec-r-pXen (n=54), and *Xenopus* (n=33).

- For context, state clearly at which IPTG concentrations the first experiments were done.

Response:

Yes, we apologize that we omitted labeling of the IPTG concentration (400 μM) in Fig. 1, and did not clearly state it in its legend and in related *Results* section (“*In vivo* assembly of the eukaryotic nucleosome core in *E. coli*”), as we were preoccupied with its description (400 μM IPTG) in *Methods* and in the legend of Supplementary Fig. 1. We are grateful for your pointing out this omission in our manuscript! Thank you very much!

Our study was designed logically into three parts. Without clear description of IPTG concentration (400 μM) for the 1st part, the logic of our study cannot be easily followed:

In the 1st part, we explored whether the eukaryotic nucleosome can assemble *in vivo* in a bacterium, whose expression was induced under 400 μM IPTG, a common condition used for foreign protein expression with pET-vector system in *E. coli*. (The experiments were designed to enable histones expression in *E. coli* and *in situ* assembly of the nucleosomes *in vivo* (Figs. 1 and 2)).

In the 2nd part, we performed titration experiments on *in vivo* nucleosome formation with various concentrations of IPTG, i.e., 1, 10, 25, 50, 75, 100, 200, and 400 μM . The experiments were designed to illustrate different histone expression levels, and correlate nucleosome formation and profiles with varying IPTG concentrations (Fig. 3).

In the 3rd part, basing on the IPTG titration data, longer-term growth experiments were conducted for the nucleosome-forming strains with 1 μM IPTG treatment, a condition under which *E. coli* was found viable and had sustained growth for at least 110 divisions. At this condition, the nucleosome-forming strains were thoroughly characterized along the passages (Figs. 4, 5, and 6).

We have updated Fig. 1 with labeling 'IPTG 400 μM ' (in yellow box), and the legends of Figs. 1 and 2 (blue text). We also added description of IPTG concentration (400 μM) to the related *Results* section (page 5, lines 115, and 117; page 6, line 127; page 8, line 189; page 9, line 196; blue text). Thank you very much!

- All growth experiments were done at 1 μM , where there are no growth defects (and the competition experiment is nice). I like the IPTG titration, but not enough detail is given. However, it appears (but one really has to look for this information) that the majority of experiments were done at this low concentration, apart from the IPTG titration experiment. Fig. 3A should include time after IPTG induction.

Response:

As described in response to the last question above, the 1st part experiments were conducted with 400 μM IPTG concentration. The 2nd part experiments were titration with various concentrations of IPTG (1, 10, 25, 50, 75, 100, 200, and 400 μM), and the 3rd part (growth experiments) were done at 1 μM IPTG. We have updated figures and legends, and related texts in *Results* (see response to the last question above) to indicate the different concentrations of IPTG used in the three parts.

For the IPTG titration experiments, we have added more details (time after IPTG induction) to Fig. 3a (yellow box), its legend (blue text), and *Methods* (page 22, lines 543-544, blue text).

For the long-term growth experiments, we have added more description of the IPTG concentration (1 μM) to Fig. 4a (yellow box), the legends of Figs. 4, 5, and 6 (blue text), and to the related *Results* section (page 11, line 265, blue text). Thank you very much!

(Note: The longer-term growth condition (1 μM IPTG treatment) was carefully chosen to allow continued cell divisions and growth. At this condition, we observed some effects from this level of nucleosome formation on *E. coli* cells. The long-term viability of the nucleosome-forming *E. coli* under the conditions likely reflected a delicate balance between cell growth and the burden of the nucleosomes.)

- It would be interesting to describe what the effect on growth is of higher IPTG concentrations. Do they die eventually? grow slower?

Response:

All data from our preliminary experiments indicate that the nucleosome-forming *E. coli* did not grow and looked abnormally at higher IPTG concentrations.

In the titration experiments using liquid culture, upon induction of histones expression with various concentrations of IPTG (1, 10, 25, 50, 75, 100, 200, and 400 μM) for 1 hour, *E. coli* cells were harvested for eCMNase assay to detect *in situ* nucleosome assembly *in vivo*. For a preliminary experiment, we continued observation on the left-over liquid cultures, and found those with the higher IPTG levels (e.g., 10 μM and

up) did not grow over next a few hours period. We looked at some cells, e.g., from 10 μM IPTG treatment - some were elongated, and formed bubble at cell ends - separation of cytoplasmic membrane from the outer membrane at cell poles, a sign of stress (see Rsp Fig. 4. below).

Rsp Fig. 4. Ec-r-pXen cells (treated with 10 μM IPTG in LB medium) observed under phase contrast microscope. Bar length = 5 μm .

To do a clean test whether the nucleosome-forming *E. coli* can be viable, we run a preliminary experiment using solid LB media with various concentrations of IPTG (1, 10, 25, 50, 75, 100, 200, and 400 μM). Using freshly transformed Ec-r-pXen cells seeded on LB plates, we only observed colony formation on 1 μM IPTG plates as well as on no-IPTG control plates the next day after overnight culture at 37 °C. We have added more details of the preliminary test to the new submission (page 11, lines 261, and 262-264, blue text).

For this manuscript, we focused on the viable nucleosome-forming *E. coli* (at 1 μM IPTG) and characterization of nucleosome formation *in vivo*. (The work has been a long-running project starting in 2018). Please allow us to describe the effect of higher level IPTG on the *E. coli* cells - a set of very complex questions, in a full study report in the future, thank you very much!

- Entire gels for MNase gels should be shown (at least in sup figs). The author's argumentation that because it is the 'same' it can be cut off is not valid, especially when all the action is very close to where the gel is cut off (e.g. 3c).

Response:

Basing on the suggestion, we have included the entire gels for all MNase results in a new supplementary figure, Supplementary Fig. 5 (Supplementary Information, pages 6-8). We have added notes in the legends of Figs. 1d, 1h, 2c, and 3c (blue text), and in the legends of Supplementary Figs. 1d and 3 (Supplementary Information, pages 2 and 4, blue text). Thank you very much!

- Fix grammar in line 130 (as efficient)

Response:

We have revised the texts to “the assembly of nucleosome complexes was more efficient than at 18 °C” (page 6, line 133, blue text). Thank you very much!

- Discussion: 'low' was changed to 'moderate' but levels are the same.

Response:

Please see our response to the question “Calling low levels of histone expression ‘moderate’ doesn’t change things. The histone levels that allow viability are very low.” (the third question for abstract).

In summary, the levels of histones expression are in the top 12.74 percentile in *E. coli* transcriptome, and in the top 9.65 to 38.71 percentile range in *E. coli* proteome. Nucleosome formation at 1 μ M IPTG is about 13.2% of that at 50 μ M IPTG, and 7.4% of that at 400 μ M IPTG. (Notably at 400 μ M IPTG level, high over-expression of foreign proteins is often induced using the pET-vector system in BL21 cells^{1,2}). So, we conclude that ‘moderate’ is appropriate basing on the estimation from the early discussion -- We thus stay with using “moderate” to describe the nucleosome-formation in the longer-term growth experiments .

We are thankful for the comment, which led us to rethink and reflect on our data and results.

- Discussion line 397 – overstated

Response:

We have made changes from “Our results showed that bacterial chromosomes can be folded *in vivo* to form nucleosome core complexes with a degree of resemblance to those found in eukaryotic hosts.”

to

“Our results showed that bacterial chromosome DNA and eukaryotic histones can assemble *in vivo* to form nucleosome core complexes with a degree of resemblance to those found in eukaryotic hosts.” (page 17, lines 407-408, blue text) Thank you!

- Line 414 – overstated ('low' nucleosome).

Response:

We have made changes from “The eukaryotic nucleosome-forming *E. coli* are viable and exhibit sustained cell division in long-term growth experiments under a condition that moderate histones expression was induced at 1 μ M IPTG.”

to

“The eukaryotic nucleosome-forming *E. coli* are viable and exhibit sustained cell division in long-term growth experiments under a condition that histones expression was induced at 1 μ M IPTG.” (page 18, lines 424-426, blue text). Thank you!

References for point-by-point response

1. Shim, Y., Duan, M.R., Chen, X.J., Smerdon, M.J. & Min, J.H. Polycistronic coexpression and nondenaturing purification of histone octamers. *Analytical Biochemistry* **427**, 190-192 (2012).
2. Speer, S.L., Guseman, A.J., Patteson, J.B., Ehrmann, B.M. & Pielak, G.J. Controlling and quantifying protein concentration in Escherichia coli. *Protein Sci* **28**, 1307-1311 (2019).
3. Farzaneh, F. & Pearson, C.K. A method for isolating uncontaminated nuclei from all stages of developing *Xenopus laevis* embryos. *J Embryol Exp Morphol* **48**, 101-8 (1978).
4. Ladoux, B. et al. Fast kinetics of chromatin assembly revealed by single-molecule videomicroscopy and scanning force microscopy. *Proc Natl Acad Sci U S A* **97**, 14251-6 (2000).
5. Basak, R. et al. Internal Motion of Chromatin Fibers Is Governed by Dynamics of Uncompressed Linker Strands. *Biophys J* **119**, 2326-2334 (2020).
6. Bustamante, C. et al. Circular DNA molecules imaged in air by scanning force microscopy. *Biochemistry* **31**, 22-6 (1992).
7. Vesenka, J. et al. Substrate preparation for reliable imaging of DNA molecules with the scanning force microscope. *Ultramicroscopy* **42-44 (Pt B)**, 1243-9 (1992).